# In-Context Learning with Unpaired Clips for Instruction-based Video Editing

## Abstract

Despite the rapid progress of instruction-based image editing, its extension to video remains underexplored, primarily due to the prohibitive cost and complexity of constructing large-scale paired video editing datasets. To address this challenge, we introduce a low-cost pretraining strategy for instruction-based video editing that leverages in-context learning from unpaired video clips. We show that pretraining a foundation video generation model with this strategy endows it with general editing capabilities, such as adding, replacing, or deleting operations, according to input editing instructions. The pretrained model can then be efficiently refined with a small amount of high-quality paired editing data. Built upon HunyuanVideoT2V, our framework first pretrains on approximately 1M real video clips to learn basic editing concepts, and subsequently fine-tunes on fewer than 150k curated editing pairs to extend more editing tasks and improve the editing quality. Comparative experiments show that our method surpasses existing instruction-based video editing approaches in both instruction alignment and visual fidelity, achieving a 12% improvement in editing instruction following and a 15% improvement in editing quality. **Code will be open-sourced.**

## 1 Introduction

Nowadays, the field of Artificial Intelligence Generated Content (AIGC) is no longer restricted to text-based control of generated results. Various conditioning methods are introduced to enable more fine-grained regulation of content generation (Ma et al., 2025). Beyond accurately producing desired outputs under different control conditions, the editing of existing images and videos also emerges as an important research direction.

To accomplish video editing, various approaches are proposed. Training-free methods modify video content by operating on attention maps in a sophisticated manner without retraining the foundation model (Qi et al., 2023; Cong et al., 2023; Kara et al., 2024; Geyer et al., 2023; Li et al., 2024; Ku et al., 2024; Fan et al., 2024; Wang et al., 2025c; Li et al., 2025). Mask-based methods employ masks as input conditions to explicitly specify editing regions, thereby providing precise guidance for both the editing location and content (Hu et al., 2024; Jiang et al., 2025; Gao et al., 2025a). However, these approaches exhibit limitations: training-free methods often compromise generation quality, while mask-based methods require additional steps to obtain segmentation results of the targeted regions. Recently, instruction-based editing models have emerged. These methods require only textual input and the original video. In image editing, instruction-based approaches already achieve rapid progress. Representative examples include closed-source models such as ChatGPT (OpenAI, 2025) and Gemini (Google, 2025), as well as open-source models such as Qwen-Image-Edit (Wu et al., 2025a), Step1X-Edit (Liu et al., 2025), and FLUX.1 Kontext Dev (Labs et al., 2025). In contrast, in video editing, a high-quality instruction-based model remains underexplored.

A major challenge in this task is that training a high-quality model requires millions of paired editing samples. Compared with image editing datasets, video editing data are scarcer in source and more difficult to process. Real editing data can be obtained through direct shooting or rendering, but the scale of real data remains limited. Alternatively, training data can be distilled from existing editing models (Cheng et al., 2023; Wu et al., 2025b; Zi et al., 2025b), yet synthetic datasets often introduce artifacts and require substantial computational resources. The shortage of high-quality training data

therefore constitutes a critical obstacle, highlighting the need for approaches that reduce dependence on large-scale annotated video editing datasets.

To address the data challenge and achieve a high-quality model, we propose a two-stage training strategy and modify a video editing framework. The model is first pretrained on video clip data and then fine-tuned on a small number of editing pairs. Specifically, a long video segment without transitions is divided into multiple short clips, from which two clips are randomly selected as the original and the pseudo-edited video. An editing instruction is then generated based on their differences, and these video clips are used in the pretraining stage to teach the model basic editing concepts. In the subsequent supervised fine-tuning (SFT) stage, a small amount of synthetic editing data is used to enhance the model's editing capability further. For the model architecture, we employ the input manner where the tokens of the original video are concatenated with the noised tokens alone sequence. To adapt to the noiseless property of the original video, the timesteps corresponding to its tokens are set to 0. Overall, with this strategy, our model relies on about 1M video clips and fewer than 150k multi-task editing pairs, yet achieves superior performance compared with models trained on several million editing data.

In the experiments, we first compare our model with existing instruction-based video editing methods. The comparisons are evaluated along two dimensions: instruction following capability and visual quality of the generated videos. In both aspects, our model achieves state-of-the-art performance. In the ablation study, we assess different training strategies, and the results show that the proposed approach allows the model to acquire basic editing capabilities during video clip pretraining and to reach superior performance after SFT. The contributions are summarized as follows:

- A two-stage training strategy is proposed, consisting of video clip pretraining for basic editing capabilities generalization and SFT on editing data to extend editing types and improve the quality of generated video.

- An instruction-based video editing model is developed, which achieves superior performance over existing video editing approaches with approximately 12% improvement in the editing instruction following and 15% in the editing quality.

## 2 RELATED WORK

**Training-free Video Editing.** Training-free video editing methods are primarily categorized into two paradigms. The first is the inversion-based approach, which follows an invert-then-denoise pipeline. FateZero (Qi et al., 2023) constrains the randomness of diffusion models to enable consistent editing. FLATTEN (Cong et al., 2023) integrates optical flow into the attention module to improve visual consistency. RAVE (Kara et al., 2024) employs a noise-shuffling strategy to achieve temporally consistent and memory-efficient editing. TokenFlow (Geyer et al., 2023) propagates intermediate features across frames to enforce temporal consistency while preserving spatial layout and motion. VidToMe (Li et al., 2024) applies a token-merging strategy that compresses self-attention tokens for temporal coherence. AnyV2V (Ku et al., 2024) leverages the edited first frame and the features of an image-to-video (I2V) model to guide the edited sequence. Videoshop (Fan et al., 2024) adopts the modified first frame and propagates the prior through latent inversion with noise extrapolation. VideoDirector (Wang et al., 2025c) introduces spatial-temporal decoupled guidance and multi-frame null-text optimization to enhance pivotal inversion. However, inversion approximation errors substantially degrade editing fidelity. An alternative is the inversion-free paradigm, which directly maps an Ordinary Differential Equation (ODE) path between the original and edited distributions. FlowEdit (Kulikov et al., 2024) proposes an inversion-free, optimization-free, and model-agnostic text-based editing framework, which is further applied to HunyuanLoom (HunyuanLoom, 2025) for video editing. FlowDirector (Li et al., 2025) models editing as an ODE-guided evolution in data space, enhanced with guidance strategies to improve semantic alignment. Although these training-free methods exploit the prior knowledge of foundation video generative models without additional training, their performance remains limited due to the inherent gap between generation and editing tasks.

**Training-based Video Editing.** To overcome the limitations of training-free methods, several training-based approaches have been introduced. Tune-A-Video (Wu et al., 2023) presents a one-shot video tuning framework that adapts a text-to-image model with spatio-temporal attention for

Figure 1: Pipeline of training data curation. (a) shows the pipeline for curating clip data from raw videos; (b) illustrates the pipeline for synthesizing and filtering editing data.

single-video editing. Video-P2P (Liu et al., 2024) tunes a text-to-set model through an approximate inversion strategy. These one-shot methods improve temporal consistency, but the optimization process for each video remains time-consuming. For more general video editing, mask-based methods are developed, supported by the construction of large-scale mask-based video editing datasets. VIVID-10M (Hu et al., 2024) introduces a hybrid image–video dataset and proposes a versatile interactive model with keyframe-guided propagation. VACE (Jiang et al., 2025) builds on Wan Video (Wan et al., 2025) and supports multiple input conditions through Context Adapter Tuning. LoRA-Edit (Gao et al., 2025a) proposes a mask-based LoRA tuning approach for region-specific video editing. To provide more convenient editing, instruction-based methods have been proposed. InstructVid2Vid (Qin et al., 2024) introduces a data pipeline for generating paired training data. InsV2V (Cheng et al., 2023) constructs a dataset with more than 400K synthetic editing pairs and trains an editing model on it. EffiVED (Zhang et al., 2024) proposes an automatic pipeline for constructing editing data from image datasets and real-world source videos. FlowV2V (Wang et al., 2025b) reformulates video editing as flow-driven I2V generation by combining first-frame editing with pseudo flow simulation. Lucy Edit (DecartAI, 2025) concatenates the original video with noise along the channel dimension, thereby reducing the token context length. InstructVEdit (Zhang et al., 2025) introduces a full-cycle instruction-based video editing approach. InsViE-1M (Wu et al., 2025b) provides a large-scale, high-quality dataset with 1M editing pairs and develops a multi-task video editing model. Señorita-2M (Zi et al., 2025b) constructs a dataset of 2M high-quality editing pairs using state-of-the-art specialized editing models. Most of these works rely on distilling existing video editing models or introducing additional control conditions to construct synthetic data. However, training exclusively on synthetic data inevitably introduces artifacts and an "AI vibe." Moreover, generating large-scale synthetic datasets requires substantial computational resources for both inference and data cleaning.

## 3 METHOD

### 3.1 DATA CURATION

**Pretraining Clip Data.** Our approach is inspired by the data pre-processing pipelines employed in foundation video generation models (Kong et al., 2024; Wan et al., 2025; Gao et al., 2025b). In these pipelines, raw videos are first segmented using a scene detection method (SceneDetect, 2025), producing multiple scene-level segments. After ensuring that no transitions appear within each segment, these segments are further divided into clips of fixed duration according to the target number of frames required by the foundation model. In the concept of video editing, an original video is provided as input, with specific parts modified while surrounding scenes, characters, and objects are preserved. Similarly, clips extracted from the same scene generally share highly similar visual content, such as the same backgrounds, characters, and objects. Since they are sampled from different temporal intervals, variations naturally occur, including camera motion, changes in character positions, and object movement. By treating one clip as the original video and another as the pseudo-edited video, the pair can be used as a training sample, where the transformation from the original to the pseudo-edited clip is regarded as a form of video editing.

Building on this idea, we adapt the data curation pipeline of foundation models for editing data collection. As illustrated in Figure 1(a), after applying basic filters on duration, resolution, and frame rate, raw videos are divided into scene-level segments. Optical flow is then computed for each segment to filter out those with low motion amplitude. The remaining segments are subsequently divided into non-overlapping clips along the timeline according to the required duration. From each segment, two clips are randomly selected as the original and pseudo-edited videos and are annotated by Step3 (Wang et al., 2025a). The annotation process generates an instruction that describes the transformation from the original clip to the pseudo-edited one. To further enrich the data, two forms of augmentation are applied: (1) rewriting action verbs in the editing instructions, such as replacing

Figure 2: In-context Instruction-based Video Editing Model (ICVE). The instruction, original video, and edited video are injected into the model via token sequence concatenation. The timesteps corresponding to the original video tokens are fixed at $0$, while the timesteps of the edited video tokens and text tokens are set to $T$.

"replace" with synonyms like "change" or "modify"; and (2) filtering out trivial instructions that involve meaningless modifications such as "brightness," "contrast," or "saturation." Through this pipeline, large-scale video clip pairs are curated for pretraining.

**SFT Editing Data.** Publicly available video editing training data remain scarce, particularly with the emergence of advanced foundation models that require high resolution, high frame counts, and high frame rates. The quality of existing datasets often fails to meet the requirements of these latest video models. To construct high-quality editing data, we design a synthetic data pipeline.

Specifically, our pipeline employs VACE (Jiang et al., 2025) as the basic video inpainting model. As illustrated in Figure 1(b), video clips are first filtered using Step3 (Wang et al., 2025a), which labels persons and objects in the foreground as well as elements in the background. Objects that are difficult to describe or unsuitable for editing are removed. GroundedSAM2 (Ren et al., 2024; Ravi et al., 2024) is then used to obtain segmentation masks. Based on these masks, a threshold is applied to exclude unsuitable targets. In particular, edited instances are required not to exceed $50\%$ of the frame area in at least $80\%$ of the frames. This constraint prevents large discrepancies between the edited and original videos, thereby ensuring stable training. Next, the mask boundaries are randomly expanded by 20–50 pixels to prevent the inpainted regions from maintaining identical shapes and sizes to the original instances. The masked regions in the original video are then set to black, and Step3 generates a caption based on the unmasked regions. Finally, the mask, the masked video, and the caption are fed into VACE (Jiang et al., 2025) to perform inpainting, producing the edited video. The original and edited videos are then annotated again by Step3 to generate an editing instruction.

After the raw editing data are generated, a multi-stage filtering process is applied to improve quality. First, videos containing large black silhouette regions, which indicate inpainting failure, are discarded. Second, Qwen2.5-VL (Bai et al., 2025b) evaluates the remaining training data along two dimensions: the accuracy of the editing instruction and the visual quality of the edited video, each scored on a 1–5 scale. Since VACE does not generate stylization editing data, style-related samples are additionally collected from InsViE-1M (Wu et al., 2025b) and Señorita-2M (Zi et al., 2025b). For the SFT stage, only samples with a score of 5 in both dimensions are retained. Finally, the amount of data from different editing types is balanced to avoid bias.

### 3.2 IN-CONTEXT INSTRUCTION-BASED VIDEO EDITING MODEL

The proposed method adopts HunyuanVideoT2V (Kong et al., 2024) as the foundation video generation model. To adapt the base model to the instruction-based video editing task, several structural modifications are introduced. Unlike text-to-video (T2V) generation, the input prompts here are editing instructions rather than textual descriptions of video content. Therefore, the prompt prefixes used in the text encoder for enriching detailed descriptions are removed, and the editing instructions are directly fed into the text encoder without additional prefixes.

In addition to the instruction prompt, the original video is also required as input. As illustrated in Figure 2, the original video is injected into the network in a token sequence concatenation manner. Specifically, in the model, both the original and the noised videos are converted from latents to tokens using the same $2 \times 2 \times 1$ patchify module. The tokens of the original video and the noised

video are then concatenated along the sequence dimension, formulated as:

$$x^t_{\text{orig, noise}} = \text{Cat}[\mathbf{P}(z_{\text{orig}}), \mathbf{P}(z^t_{\text{noise}})], \tag{1}$$

where $z_{\text{orig}}$ and $z^t_{\text{noise}}$ denote the pure latents of the original video and the noised latents of the editing video, respectively. $\mathbf{P}$ is the patchify module, and Cat denotes concatenation along the sequence dimension. $x^t_{\text{orig, noise}}$ represents the total visual tokens for the DiT block input.

Since the original video is provided without added noise, the timesteps corresponding to the original video tokens are fixed to 0, simulating the noise-free case. The text tokens and noised tokens retain their original timesteps in the range of 0–1. These distinct timesteps affect the modulation operation in the DiT block, which predicts different scale, shift, and gate parameters according to the input timesteps. This process is formulated as:

$$x^t_{\text{text, orig, noise}} = \text{Cat}[\mathbf{M}(x_{\text{text}}, t = T), \mathbf{M}(x_{\text{orig}}, t = 0), \mathbf{M}(x^t_{\text{noise}}, t = T)], \tag{2}$$

where $\mathbf{M}$ denotes the modulation function, including scale, shift, and gate. $x_{\text{text}}$, $x_{\text{orig}}$, and $x^t_{\text{noise}}$ represent the text tokens, original video tokens, and noised edited video tokens, respectively, and $t$ indicates the input timesteps.

With this design, the DiT block treats the original video tokens as noise-free tokens with a timestep of 0, thereby preserving the visual information of the original video and ensuring high-quality generation. Moreover, by applying distinct scale, shift, and gate parameters to the original and noised tokens, the distributions of the two token types remain separated in the attention calculation, enabling the model to distinguish between them effectively. After these modifications, our model directly takes the original video and the instruction prompt as inputs. The noised tokens inherit visual content from the original video and modify specific elements under the guidance of the instruction prompt, thereby accomplishing instruction-based video editing.

### 3.3 TRAINING DETAILS

**Objective Function.** During training, the editing instruction prompt $y$, the original video $V_{\text{orig}}$, and the edited video $V_{\text{edit}}$ are provided. Both videos are first encoded by the VAE $\mathcal{E}$ to produce video latents $z_{\text{orig}} = \mathcal{E}(V_{\text{orig}})$ and $z_{\text{edit}} = \mathcal{E}(V_{\text{edit}})$. The editing instruction is fed into the text encoder to generate text tokens $x_{\text{text}}$. Noise $\epsilon$ is added to the edited latent $z_{\text{edit}}$ to obtain the noised latent $z^t_{\text{noise}}$ over timesteps $t \in (0, 1)$. The video editing model then predicts the noise $\epsilon$ added to $z_{\text{edit}}$, conditioned on $t$, $z_{\text{orig}}$, and $x_{\text{text}}$. The training objective is defined as:

$$\mathcal{L} = \mathbb{E}_{\epsilon \sim \mathcal{N}(0,1),\, t}\left[\left\|\epsilon - \epsilon_\theta(t, z^t_{\text{noise}}, z_{\text{orig}}, x_{\text{text}})\right\|^2_2\right], \tag{3}$$

where $\epsilon_\theta$ denotes the noise predicted by the video editing model parameterized by $\theta$.

**Pretraining on Clip Data.** In the pretraining stage, the training process relies solely on video clips without using any editing data. Pretraining on clips offers several advantages. First, existing data pre-processing pipelines from foundation video models (Wan et al., 2025; Kong et al., 2024; Gao et al., 2025b) can be directly reused for data collection. These pipelines are simple and mature, enabling the acquisition of large volumes of usable data. Second, video clips from real-world sources generally exhibit higher visual quality than synthetic editing data, as they are free from artifacts, "AI vibe," or other implausible elements. Video clips extracted from the same scene segment help the model learn to preserve contextual information of the original video, including scene layout, character identity, and object appearance. This strengthens the model's ability to retain original video content during editing operations. Meanwhile, clips sampled from different temporal intervals introduce motion variations, which can be regarded as a form of video-to-video editing. Although such clip data are not strictly aligned editing pairs along the same timeline, they still allow the model to learn basic editing concepts from temporal differences. Even with pretraining only on video clips, the model acquires initial editing capabilities, as later demonstrated in the ablation study.

Specifically, pretraining begins at 240p resolution, leveraging large-scale video clips to guide the model in extracting and preserving scene, character, and object information from the original video input. At this stage, the model also develops a basic ability to follow simple editing instructions, such as addition, removal, and replacement operations. The resolution is then progressively increased from 240p to multiple resolution buckets to further enhance visual quality. Overall, this pretraining stage consumes approximately 1M video clip data.

Table 1: Comparison results with instruction-based video editing methods. Best results in each column are highlighted in **bold**, and the second best are underlined.

| Method | Extra Model | Editing Instruction | | | | | | Video Quality | | | |
|---|---|---|---|---|---|---|---|---|---|---|---|
| | | CLIP | Pick | I_F | O_P | E_Q | S_R | S_C | B_C | M_S | T_F |
| AnyV2V (Ku et al., 2024) | +InsP2P | 0.2534 | 19.6328 | 2.5896 | 3.2428 | 3.3526 | 25.43% | 0.9405 | 0.9509 | 0.9826 | 0.9750 |
| Videoshop (Fan et al., 2024) | | 0.2425 | 19.3602 | 2.5260 | 2.5549 | 2.5549 | 23.70% | 0.9449 | 0.9535 | 0.9742 | 0.9550 |
| Señorita-2M (Zi et al., 2025b) | | 0.2443 | 19.5202 | 2.5954 | 2.9191 | 3.0058 | 23.12% | 0.9480 | 0.9616 | 0.9910 | 0.9813 |
| AnyV2V (Ku et al., 2024) | +Qwen | 0.2678 | 20.0900 | 3.8439 | 3.9422 | 3.3526 | 64.16% | 0.9088 | 0.9292 | 0.9795 | 0.9719 |
| Videoshop (Fan et al., 2024) | | 0.2605 | 19.7915 | 3.2890 | 3.4046 | 2.9075 | 37.57% | 0.9273 | 0.9419 | 0.9752 | 0.9571 |
| Señorita-2M (Zi et al., 2025b) | | **0.2729** | **20.4642** | 4.0549 | 4.0809 | 3.5260 | **78.03%** | 0.9601 | 0.9651 | 0.9919 | 0.9827 |
| InsV2V (Cheng et al., 2023) | - | 0.2658 | 19.7429 | 2.3988 | 4.0694 | 3.4046 | 30.06% | 0.9630 | **0.9708** | 0.9873 | 0.9779 |
| InsViE-1M (Wu et al., 2025b) | | 0.2489 | 19.5580 | 1.8902 | 3.2312 | 2.5376 | 18.50% | 0.9675 | 0.9565 | 0.9810 | 0.9572 |
| Lucy Edit Dev (DecartAI, 2025) | | 0.2531 | 19.8444 | 2.2543 | 4.1387 | 3.6474 | 24.28% | 0.9668 | 0.9459 | 0.9922 | 0.9838 |
| Ours | | 0.2701 | 20.2516 | **4.5491** | **4.3064** | **4.2081** | **78.03%** | **0.9795** | 0.9636 | **0.9941** | **0.9867** |

**SFT on Editing Data.** After pretraining, the model is capable of preserving the content of the original video and demonstrates basic editing abilities. However, clip data does not provide strict editing concepts along the timeline and cannot represent stylization editing types. Therefore, SFT on high-quality editing data is required.

Following prior strategies for foundation models (Wan et al., 2025; Kong et al., 2024; Gao et al., 2025b), it is observed that SFT requires only a small amount of data and limited training time. Our SFT process adopts fewer than 150k high-quality editing video pairs and is conducted for one epoch. This setup strengthens the model's editing capabilities while avoiding collapse and reducing the risk of overfitting caused by the relatively small dataset size. After SFT, the model learns to respond precisely to editing instructions and to generate high-quality editing results. It also enhances the ability to preserve fine-grained details in regions that remain unchanged. Moreover, the model adapts from supporting limited editing types after pretraining to handling a wide range of editing tasks after SFT. In summary, approximately 75% of the training is dedicated to pretraining on video clip data, while less than 25% involves high-quality editing data for SFT.

## 4 EXPERIMENTS

### 4.1 EXPERIMENTAL SETTINGS

**Compared Methods.** We compare our model with both training-free and training-based video editing methods that accept instruction input. The training-free methods include AnyV2V (Ku et al., 2024) and Videoshop (Fan et al., 2024), while the training-based methods include InsV2V (Cheng et al., 2023), Señorita-2M (Zi et al., 2025b), InsViE-1M (Wu et al., 2025b), and Lucy Edit Dev (DecartAI, 2025). Since AnyV2V, Videoshop, and Señorita-2M require the first edited frame as an additional input, results are reported with the aid of two instruction-based image editing models: InstructPix2Pix (+InsP2P) (Brooks et al., 2023) and Qwen-Image-Edit (+Qwen) (Wu et al., 2025a).

**Testing Dataset.** Following previous works (Fan et al., 2024; Zi et al., 2025b; Wu et al., 2025b), we construct a test set containing 300 video samples from the DAVIS (Pont-Tuset et al., 2017) and YouTubeVOS (Xu et al., 2018) datasets, as well as videos from the Pexels website (Pexels, 2025). GPT-5 (OpenAI, 2025) is employed to generate diverse editing instructions for these original videos.

**Evaluation Metrics.** Evaluation is conducted from two perspectives: instruction following and video generation quality. For instruction following, CLIP text–image embedding similarity (Radford et al., 2021) and the Pick score (Kirstain et al., 2023) are employed. Since these metrics cannot fully capture editing effectiveness, GPT-5 (OpenAI, 2025) is additionally adopted for automated evaluation. GPT-5 assesses each test sample along four dimensions: alignment between the instruction and the edited video (Instruction Following, **I_F**), preservation of unedited regions from the original video (Original Video Preservation, **O_P**), overall editing quality (Editing Quality, **E_Q**), and success ratio of edits (Success Ratio, **S_R**). Scores range from 1 to 5, and the reported results are averaged across all test samples. For video generation quality, we adopt the evaluation metrics from VBench (Huang et al., 2024). The generated videos are evaluated on Subject Consistency (**S_C**), Background Consistency (**B_C**), Motion Smoothness (**M_S**), and Temporal Flickering (**T_F**).

## 4.2 COMPARISON WITH SOTA MODELS

As shown in Table 1, our method achieves the best performance on most metrics. For instruction-related metrics, it obtains the highest scores in Instruction Following, Original Video Preservation, Editing Quality, and Success Ratio, while ranking second in the CLIP and Pick scores. Our model achieves a 12% improvement in editing instruction following and a 15% improvement in editing quality. Although Señorita-2M+Qwen-Image-Edit (Zi et al., 2025b; Wu et al., 2025a) achieves the same Success Ratio, its Editing Quality score is lower than ours. This suggests that it fails to effectively propagate prior information from the first edited frame to subsequent frames. Consequently, both its Instruction Following and Editing Quality scores remain inferior to ours. Lucy Edit Dev (DecartAI, 2025), which is based on Wan Video (Wan et al., 2025), achieves good performance on Original Video Preservation and Editing Quality scores, but underperforms on Success Ratio.

Regarding video generation quality, our method achieves the best results in Subject Consistency, Motion Smoothness, and Temporal Flickering, benefiting from the strong foundation model and the high-quality curated data used for training. The Background Consistency metric is also competitive, ranking third among all methods. Qualitative comparisons in Figure 4 further demonstrate that our method produces more precise and natural editing results than other instruction-based approaches.

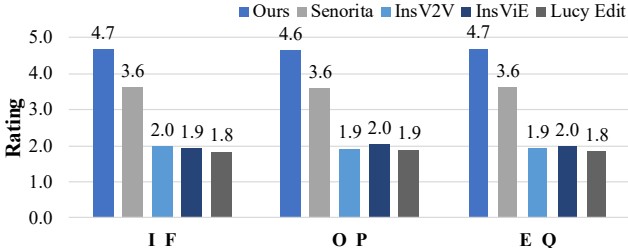

Figure 3: User study results of instruction-based video editing models.

## 4.3 USER STUDY

We collect 20 videos covering diverse editing types for the user study. Edited videos are generated using the official codes and weights of Señorita-2M (Zi et al., 2025b), InsV2V (Cheng et al., 2023), InsViE-1M (Wu et al., 2025b), and Lucy Edit Dev (DecartAI, 2025). The user study is designed to be completed within 10–20 minutes. For each case, the interface presents the videos produced by the five methods listed above, and participants are instructed to rate the videos along three evaluation dimensions: (i) "Instruction Following"; (ii) "Original Video Preservation"; and (iii) "Editing Quality". In total, 50 valid questionnaires are collected. Each evaluation metric is computed as the average rating over all $50 \times 20$ individual judgments.

As illustrated in Figure 3, our method achieves the highest scores across all evaluation dimensions. These results demonstrate that our approach produces high-quality edited videos and delivers superior performance in instruction following, original video preservation, and overall editing quality.

## 4.4 ABLATION

**Ablation on Data Scale and Training Steps.** We investigate the impact of different amounts of clip data and SFT data on training. In the pretraining stage, the batch size is set to 32. As shown in Figure 5, after 40K training steps (corresponding to 1.28M clips, evaluated every 10K steps), the model learns to preserve original video content and maintain good video quality, but it cannot yet respond precisely to editing instructions. Some edited results generated by the pretrained model are shown in Figure 6(c). These results indicate that the model has already acquired some basic editing concepts from the video clips. However, due to the absence of editing data during training, the pretrained model cannot generate results that strictly follow the timeline of the original videos.

To analyze the impact of pretraining data scale, we conduct an ablation study where approximately 100K clips (10% of the original amount) are selected for pretraining. As shown in Figure 6(a), insufficient pretraining leads to noticeable degradation in generation quality, resulting in severe artifacts and unnatural structures, such as a duck with two heads or a woman with multiple legs. Since our editing model introduces structural modifications on top of the foundation model, inadequate

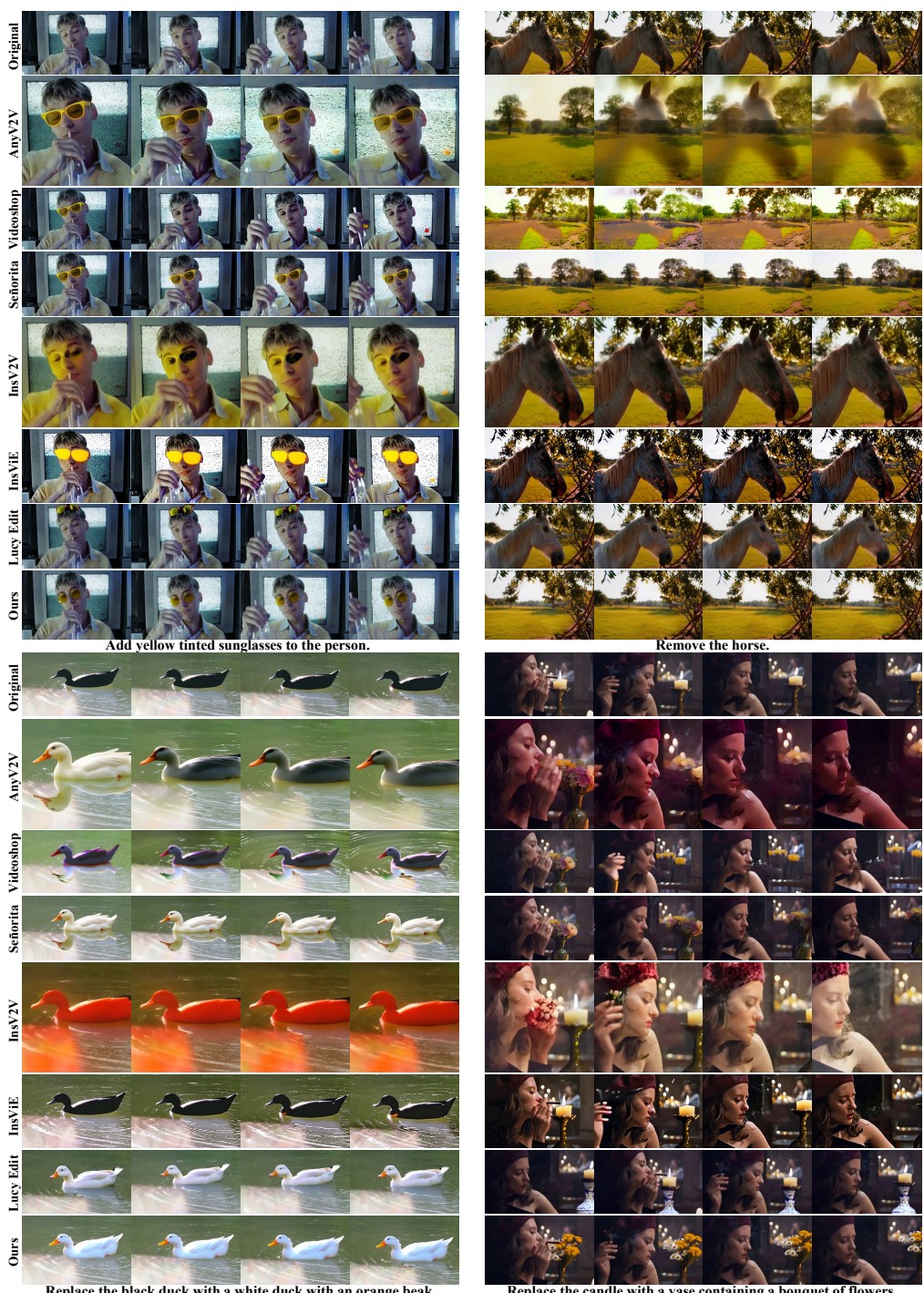

Figure 4: Comparison results with the instruction-based methods. For methods that require the first edited frame, the first frame is obtained using Qwen-Image-Edit (Wu et al., 2025a).

pretraining prevents the model from adapting to these changes, thereby impairing the underlying generative capability. Therefore, sufficient pretraining steps and a large quantity of pretraining data are essential, as they allow the modified model to recover the generation quality of the foundation model. Our method addresses the challenge of acquiring large-scale high-quality pretraining data by constructing video clip pairs, which allows the substantial data requirements to be met efficiently and at low cost. We also observe limitations caused by excessively long pretraining. Figure 6(b) presents the same test case edited by models pretrained with different numbers of steps. Notably, this case involves stylization editing, which cannot be learned from clip data. The results show that

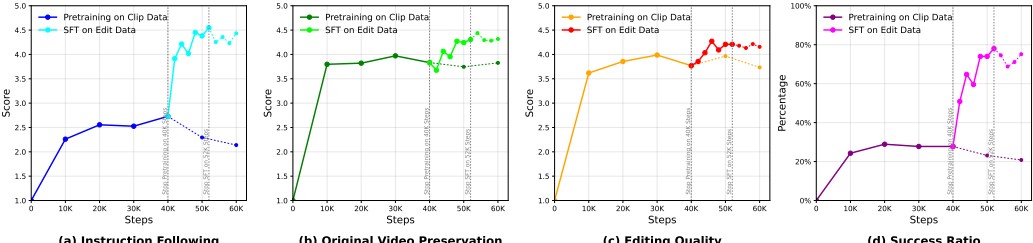

Figure 5: Model performance with varying training steps and data. Dashed lines indicate evaluation results from continued training. The initial segment of the curve in each sub-figure corresponds to the pretraining stage, where the model acquires basic editing capabilities from clip data. The dashed line after 40K steps illustrates that prolonged pretraining leads to model degradation. The second segment demonstrates that the model improves rapidly with only a few SFT steps, and the dashed line indicates that approximately 12K SFT steps are sufficient to yield a high-quality editing model.

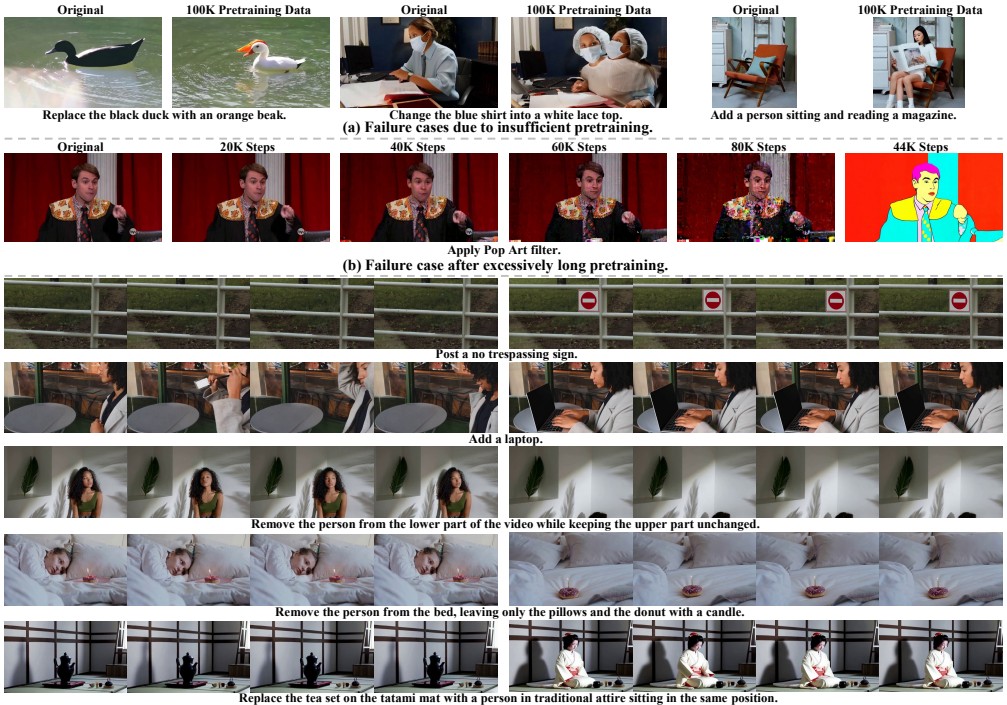

Figure 6: Visualization results of the pretrained model. (a) illustrates that insufficient pretraining leads to degradation of the model's inherent generative capability, causing the edited videos to exhibit unnatural artifacts. (b) shows that excessive pretraining leads to overfitting and collapse on unseen editing types. However, with a small number of SFT steps on editing data, the pretrained model rapidly learns new editing types. (c) demonstrates that the pretrained model acquires basic editing operations, such as addition, removal, and replacement, using only video clip data. Original videos are shown on the left, and edited results are on the right.

with prolonged pretraining (after 60K steps), the model tends to overfit to the clip data and eventually collapse. The dashed lines of pretraining in Figure 5 also validate this degradation phenomenon. Therefore, we terminate pretraining at 40K steps.

The impact of the amount of SFT data is also examined. In this stage, the batch size is set to 12. As shown in Figure 5, the model learns quickly from the SFT data. After only 4K training steps (48K editing pairs, evaluated every 2K steps), the model can perform accurate video editing. Furthermore, we observe that 120K–144K editing pairs are sufficient to train a high-quality model, indicating that only a relatively small amount of editing data is required in the SFT stage. Figure 6(a) shows the stylization editing results at 44K steps. Even though the clip data does not contain any stylization editing examples, the model acquires new editing capabilities within a few SFT steps.

**Ablation on Training Strategies.** We evaluate the effectiveness of the proposed strategy of first pretraining on video clip data and then performing SFT with a small amount of editing data. Three models are trained under different settings: (1) the model trained with the proposed strategy; (2) the model pretrained on editing data and then fine-tuned on the same SFT data, representing the standard training paradigm of editing models; (3) the model trained directly on the same editing data for SFT without a pretraining stage, serving to examine whether a small amount of editing data alone is sufficient to train a high-quality model. We visualize the evolution of the Instruction Following score during training under the different settings, and the best performance achieved by each setting is reported accordingly.

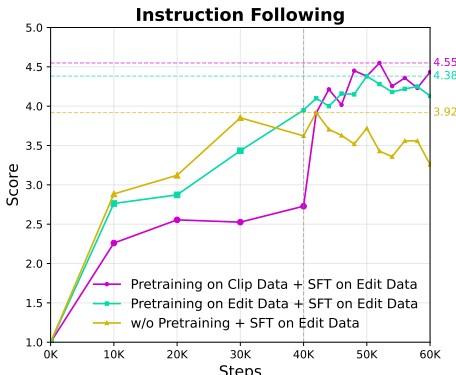

Figure 7: Instruction Following score variation on the three training settings.

As shown in Figure 7, under Setting (1), the model learns editing concepts gradually during the pretraining stage on clip data (before 40K steps). Since the pseudo editing pairs are constructed from video clips, the Instruction Following score improves only slowly at this stage. Once the SFT editing data is introduced, the instruction following ability increases rapidly, and the model reaches its best performance after approximately 10K steps of SFT training. Under Setting (2), the standard training paradigm for editing models is simulated. The model first improves its editing ability during pretraining and subsequently enhances its performance during the SFT stage. In contrast to our proposed strategy, however, this approach consumes a large amount of editing data during the initial 40K pretraining steps, and preparing such a large pretraining dataset requires substantial computational resources. Under Setting (3), only a small amount of high-quality SFT editing data is used for training. Due to the higher data quality, the score increases at the fastest rate at the beginning of training. However, the limited size of the SFT dataset causes the model to overfit, leading to model collapse. After reaching its highest score early in training, the performance subsequently degrades. As shown in Table 2, the standard training paradigm fails to produce a model of higher quality than the one obtained using our proposed strategy. This limitation primarily arises because training solely on synthetic data inevitably introduces artifacts, while the instruction labels in synthetic datasets are often imprecise, degrading the final performance. Moreover, this paradigm relies on an extremely large dataset (approximately 3M samples), and preparing such data demands considerable computational resources. Additionally, the results of setting (3) further demonstrate that training exclusively on a small amount of SFT editing data (about 150K) is insufficient, as the model overfits quickly and collapses. Together, these findings highlight both the need for the pretraining stage and the effectiveness of the proposed two-stage training strategy.

Table 2: Ablation study on training strategies.

| Setting | I_F | O_P | E_Q | S_R |
|---|---|---|---|---|
| Pretraining on Clip Data + SFT on Edit Data (Ours) | 4.5491 | 4.3064 | 4.2081 | 78.03% |
| Pretraining on Edit Data + SFT on Edit Data | 4.3815 | 4.0809 | 4.0520 | 72.25% |
| w/o Pretraining + SFT on Edit Data | 3.9191 | 3.6647 | 3.7746 | 56.65% |

## 5 CONCLUSION

In this paper, we present a data-efficient training strategy for instruction-based video editing. Our method combines pretraining on video clips with SFT on a relatively small amount of editing data. The pretraining stage generalizes the model with basic editing capabilities, and the SFT stage extends the editing type and improves video quality. We introduce a data pipeline for curating clip data and synthesizing high-quality editing data, and further adapt an in-context instruction-based video editing model. Extensive experiments demonstrate that the proposed approach significantly reduces reliance on large-scale synthetic editing datasets while achieving superior performance in both editing instruction following and video generation quality compared with existing approaches.

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

APPENDIX

## A TRAINING DETAILS.

**Pretraining Stage.** The model is initialized with the weights of HunyuanVideoT2V (Kong et al., 2024). In the 240p pretraining stage, three resolution buckets are used: (240, 384, 77), (384, 240, 77), and (256, 256, 125), where the triplets denote height, width, and number of frames, respectively. The batch size is set to 32, and the 240p pretraining is conducted for approximately 30K iterations. In the subsequent 480p pretraining stage, the resolution buckets are set to (480, 768, 77), (768, 480, 77), and (512, 512, 125). The batch size remains 32, and training continues for an additional 10K iterations. In total, 40K iterations of pretraining are performed using only video clip data, with the learning rate fixed at $1 \times 10^{-5}$. This stage consumes about 1.28M video clips.

**SFT Stage.** To accommodate the SFT editing data from multiple sources, seven resolution buckets are adopted: (480, 768, 77), (768, 480, 77), (576, 1024, 21), (1024, 576, 21), (336, 592, 29), (592, 336, 29), and (480, 480, 125). The global batch size is set to 12, and a mini-batch strategy is applied to balance the number of tokens across different buckets. SFT is conducted for 12K iterations on high-quality editing data, with the learning rate set to $1 \times 10^{-6}$. This stage consumes about 144K of editing data.

**Overall Training.** The full training process consists of 52K iterations, with pretraining accounting for roughly 75% of the total training time, and the remaining 25% dedicated to SFT on a relatively small amount of high-quality editing data. The model is fully trained throughout the entire process.

## B INFRASTRUCTURE OPTIMIZATION

To enable scalable and efficient training, we adopt several system-level optimizations. First, advanced parallelism strategies are introduced to address the challenges posed by large-scale models with long contexts. Second, a fine-grained activation checkpointing scheme is applied to reduce memory usage and better balance computation with communication. Finally, distributed checkpointing is employed to ensure efficient and scalable saving and loading of training states. The details of these optimizations are described below.

**Model Parallelism.** The large model size and extremely long sequence length necessitate multiple parallelism strategies for efficient training. We employ 3D parallelism, which scales along three dimensions: input sequences, data, and model parameters. For input sequences, Sequence Parallelism (Li et al., 2021; Korthikanti et al., 2023; Jacobs et al., 2023) shards samples across sequence-parallel groups at the start of training. During attention computation, all-to-all communication distributes query, key, and value shards so that each worker processes the full sequence but only a subset of attention heads. After parallel computation, another all-to-all step aggregates the outputs, recombining both the attention heads and the sharded sequence dimension. For parameters, gradients, and optimizer states, we use Fully Sharded Data Parallelism (Zhao et al., 2023), which applies full sharding within each shard group while replicating parameters across groups. This approach effectively implements data parallelism while reducing communication costs by limiting all-gather and reduce-scatter operations.

**Activation Checkpointing.** Selective activation checkpointing (Chen et al., 2016) is applied to minimize the number of layers requiring activation storage while maximizing GPU utilization, thereby improving training efficiency without excessive memory consumption.

**Distributed Saving and Loading.** To support scalable training, we adopt a distributed checkpointing solution (Lian et al., 2024) that enables parallel saving and loading of partitioned states with high I/O efficiency. It also supports resharding across distributed checkpoints, providing flexibility to switch seamlessly between different training scales, numbers of ranks, and storage backends.

## C COMPUTATIONAL RESOURCES STATEMENT.

**Data Curation.** For the curation of pretraining clip data, CPU nodes are primarily used to segment long videos based on transitions and to subdivide each segment into short clips. GPU resources

are utilized mainly during the annotation phase, where Step3 (Wang et al., 2025a) is deployed on 8 NVIDIA H20 GPUs to generate labels for the pretraining data. Compared to other large-scale video editing datasets, our pretraining data curation pipeline requires GPU usage only for instruction labeling and motion filtering by optical flow, thereby avoiding the computationally expensive process of generating synthetic video data. For the SFT editing data, GPU resources are primarily consumed during three steps: (i) video inpainting using VACE 1.3B (Jiang et al., 2025); (ii) editing instruction generation using Step3; and (iii) data filtering using Qwen2.5-VL (Bai et al., 2025b). All components are deployed on 8 NVIDIA H20 GPUs. Since our SFT stage requires only a relatively small amount of synthetic data (Ours: 150K vs. InsViE (Wu et al., 2025b): 1M, Señorita (Zi et al., 2025b): 2M, and Ditto (Bai et al., 2025a): 1M), the overall cost of SFT data preparation remains substantially lower than that of prior large-scale video editing pipelines.

**Model Training.** The training of the video editing model is conducted on 32 NVIDIA H100 GPUs. The pretraining stage takes approximately 6-7 days, and the SFT stage requires roughly 2-3 days.

## D    MORE COMPARISON RESULTS.

**Compared with Specialized Models.** Our model is also compared with specialized models designed for specific editing tasks. For addition and replacement tasks, VACE (Jiang et al., 2025) is selected as the specialized model, while for the removal task, Minimax-Remover (Zi et al., 2025a) is used. For video stylization, the open-source method Ditto (Bai et al., 2025a) is selected for comparison. Evaluations are conducted only on the corresponding task-specific subsets. Note that some specialized models require either a pre-calculated mask or descriptive text as input. Therefore, the editing instructions in our evaluation dataset are rewritten into descriptive text by GPT-5 (OpenAI, 2025), and bounding boxes for the regions to be edited are generated by GPT-5 based on the instructions. Corresponding segmentation masks are then produced using SAM2 (Ravi et al., 2024).

Table 3 presents the evaluation results for each task, where our method achieves superior performance across all metrics. Figure 8 illustrates the comparison between our method and specialized methods across different editing tasks. In the addition task, the target element (e.g., a hat) does not exist in the original video, which causes the segmentation model to fail to produce an accurate mask near the child's head. Consequently, the final edited output alters the child's entire appearance and clothing. In the replacement task, VACE Jiang et al. (2025) merely changes the color of the barrette to beige rather than correctly replacing it with a T-shirt. These results highlight the inherent limitations of mask-based approaches, as their outputs depend heavily on the quality and accuracy of the input masks. In the removal task, Minimax-Remover (Zi et al., 2025a) exhibits noticeable structural collapse and artifacts, which further demonstrates its lack of robustness.

Table 3: Comparison results with specialized video editing models on addition, removal, replacement, and stylization tasks. Best results in each task are highlighted in **bold**.

| Method | Task | Editing Instruction | | |
|---|---|---|---|---|
| | | I_F | O_P | E_Q |
| VACE (Ku et al., 2024) | Addition | 2.36 | 3.32 | 3.98 |
| Ours | | **4.42** | **4.58** | **4.36** |
| Minimax-Remover (Zi et al., 2025a) | Removal | 4.02 | 4.28 | 3.74 |
| Ours | | **4.11** | **4.46** | **3.87** |
| VACE (Ku et al., 2024) | Replacement | 3.79 | 4.74 | 3.98 |
| Ours | | **4.44** | **4.79** | **4.37** |
| Ditto (Bai et al., 2025a) | Stylization | 4.38 | 3.35 | 4.38 |
| Ours | | **4.81** | **3.46** | **4.61** |

**Results on Out-of-Distribution Data.** To evaluate the model's generalization capability, anime and synthetic videos are used as input, accompanied by editing instructions, to form an out-of-distribution (OOD) test set. Note that neither our pretraining data nor our SFT data contains samples of this type.

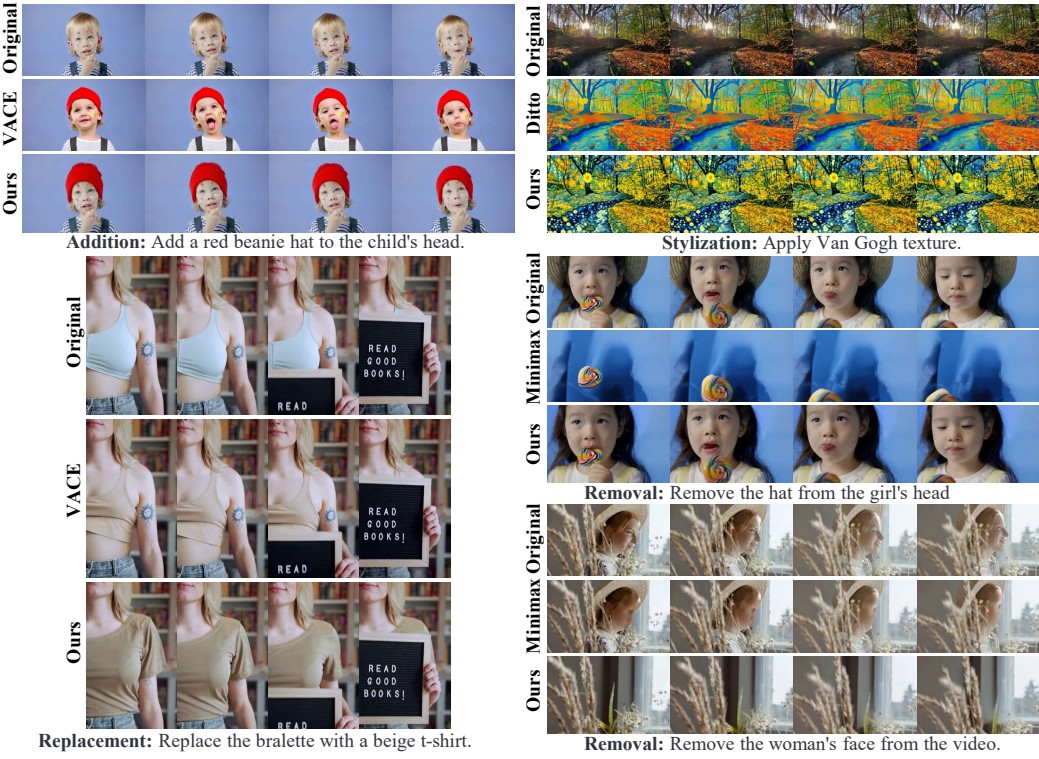

Figure 8: Comparison results with specific editing model. Addition and Replacement tasks: VACE (Ku et al., 2024); Removal task: Minimax-Remover (Zi et al., 2025a); Stylization Task: Ditto (Bai et al., 2025a).

As shown in Figures 9 and 10, our model produces reasonable editing results on OOD data. However, several failure cases are observed. For example, in Figure 9, the person's facial structure changes slightly after the mask is removed; in Figure 10 (second row), the woman's eyes become smaller following the addition of glasses; and in Figure 10 (third row), although a ring is successfully added, the generated hand appears in a photorealistic style. These limitations could be alleviated by expanding the corresponding data types during both the pretraining and SFT stages.

## E MORE ABLATION STUDIES.

**Ablation on Foundation Models.** To further verify the generalization of our training strategy and model modifications, we re-implement the video editing model on another open-source SOTA video generation foundation model, Wan 2.1 14B (Wan et al., 2025). Similarly, the same modifications are applied to the DiT blocks of Wan 2.1. We inject the original video into the model via token sequence concatenation. The timesteps of the original video tokens are fixed at 0, while the timesteps of the text tokens and noisy video tokens are kept at $T$. Using the same pretraining clip data and SFT editing data, we train the video editing model under identical strategies and computational resources.

As shown in Table 4, the model built on Wan 2.1 14B (Wan et al., 2025) achieves higher scores on editing instruction metrics compared to the model based on HunyuanVideo (Kong et al., 2024). Benefiting from the stronger foundation model, the system also exhibits improvements in several video quality metrics. Figures 11 and 12 present qualitative editing results generated based on the Wan 2.1 14B foundation model. **The new code and model are also open-sourced.**

**Ablation on Curated Data.** To verify the effectiveness of our proposed training strategy and data pipeline, an editing model is trained using only Señorita-2M (Zi et al., 2025b) with identical training steps and the same model architecture. As shown in Table 5, the video editing model trained

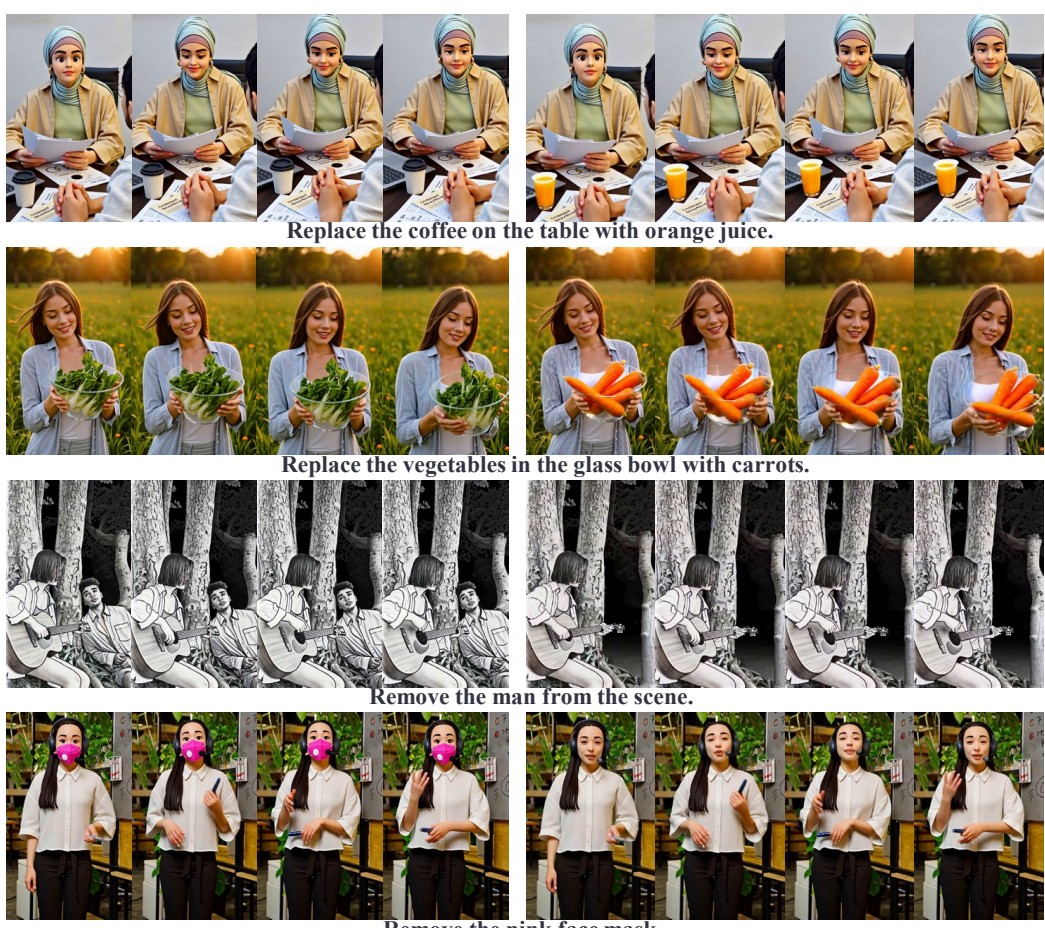

Figure 9: Editing results produced by our model on OOD test data. Original videos are shown on the left, and edited results are on the right.

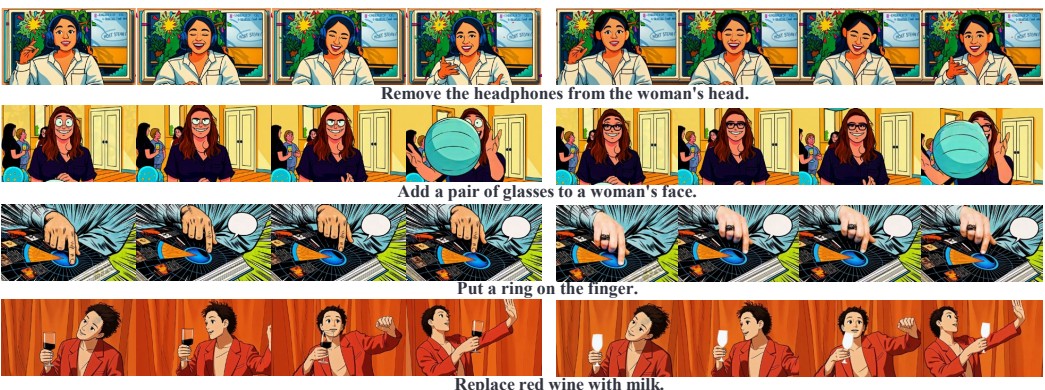

Figure 10: Editing results produced by our model on OOD test data. Original videos are shown on the left, and edited results are on the right.

on Señorita-2M (Zi et al., 2025b) is limited by the quality of its training data and therefore performs worse in both instruction following and video quality than the model trained on our curated dataset. As illustrated in Figure 13, the model trained on Señorita-2M fails to fully follow the editing instructions and exhibits noticeable unnatural artifacts.

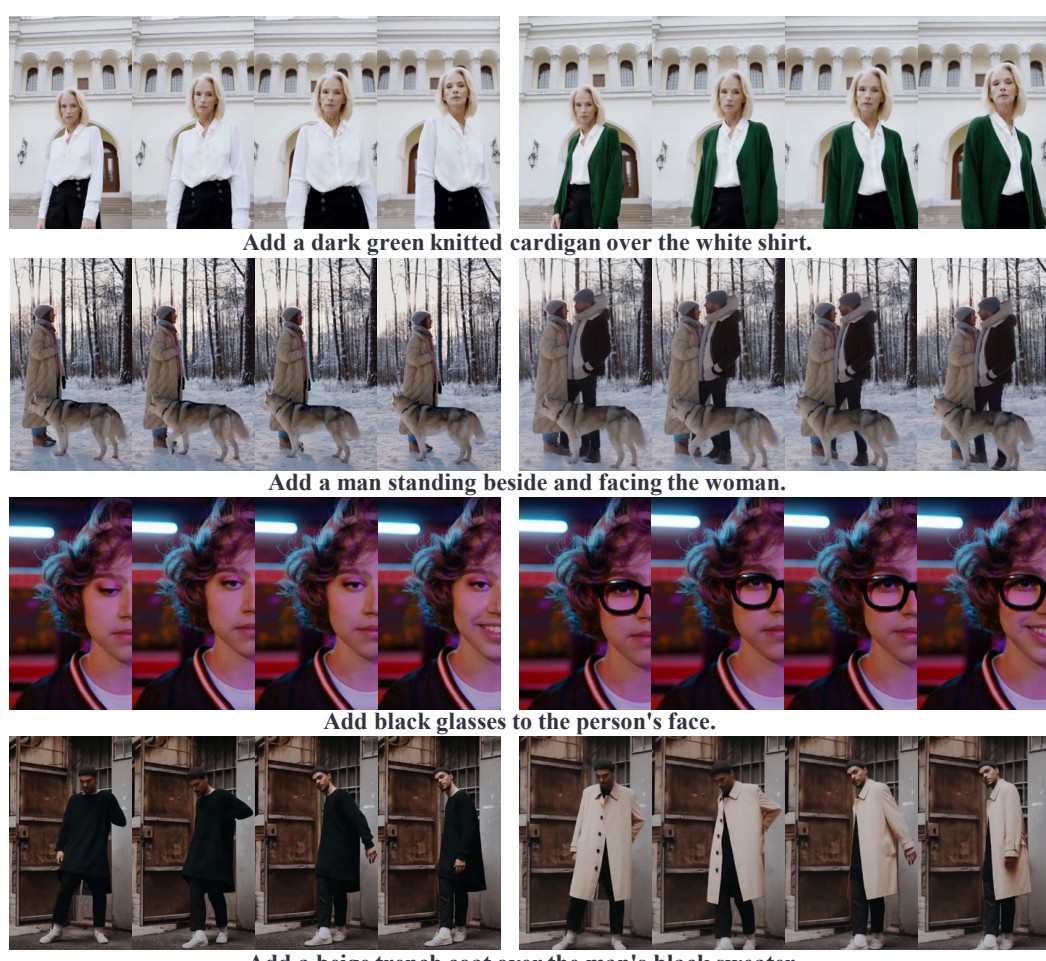

Figure 11: Editing results by our model based on Wan2.1 14B (Wan et al., 2025). The left are the original videos, and the right are the edited results.

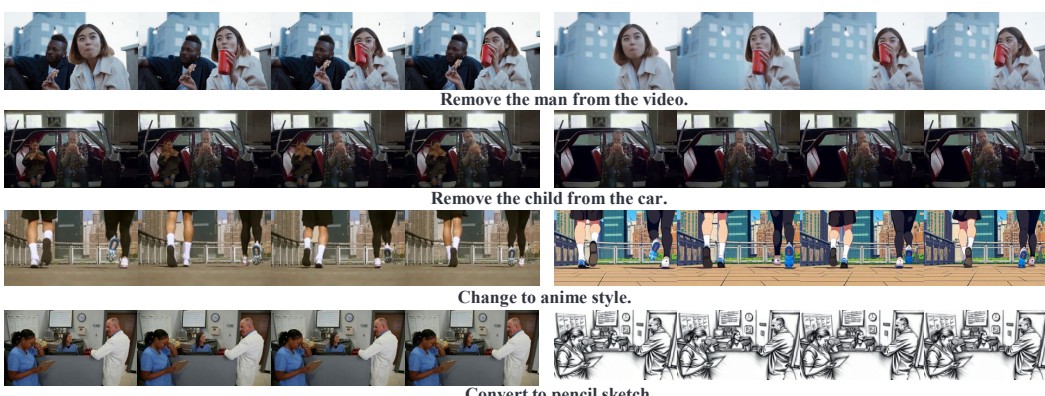

Figure 12: Editing results by our model based on Wan2.1 14B (Wan et al., 2025). The left are the original videos, and the right are the edited results.

**Ablation on Timestep for Original Video.** We conduct an ablation study on the timestep configuration by modifying the model to use the same timestep for both the original video tokens and the noisy video tokens. We observe that setting identical timesteps leads to poorer preservation of

Table 4: Ablation on foundation models.

| Foundation Model | Editing Instruction | | | | Video Quality | | | |
|---|---|---|---|---|---|---|---|---|
| | I_F | O_P | E_Q | S_R | S_C | B_C | M_S | T_F |
| HunyuanVideo (Kong et al., 2024) | 4.5491 | 4.3064 | 4.2081 | 78.03% | 0.9795 | 0.9636 | 0.9941 | 0.9867 |
| Wan2.1 14B (Wan et al., 2025) | 4.6011 | 4.3116 | 4.2405 | 80.24% | 0.9781 | 0.9562 | 0.9947 | 0.9864 |

Table 5: Ablation on training data.

| Training Data | Editing Instruction | | | | Video Quality | | | |
|---|---|---|---|---|---|---|---|---|
| | I_F | O_P | E_Q | S_R | S_C | B_C | M_S | T_F |
| Señorita-2M (Zi et al., 2025b) | 3.0578 | 3.3179 | 3.9769 | 36.99% | 0.9760 | 0.9602 | 0.9921 | 0.9823 |
| Our curated pretraining and SFT data | 4.5491 | 4.3064 | 4.2081 | 78.03% | 0.9795 | 0.9636 | 0.9941 | 0.9867 |

fine details, such as the appearance of the eyes in Figure 14 (left). This occurs because the model interprets the original video tokens as being corrupted by noise ($t = T$), causing it to discard some original information during the generation process.

We also observe that using the same timestep hinders the model's ability to distinguish between the original and the edited video. As illustrated in Figure 14 (right), the model fails to completely replace the camera with a phone and exhibits copy-paste artifacts. This suggests that the model confuses the tokens corresponding to the original video (camera) with those of the edited result (smartphone). These results demonstrate that distinct timesteps for the original ($t = 0$) and noisy ($t = T$) video tokens are essential for both the accurate preservation of the original video and reliable editing results.

**Performance of Original Video Preservation.** We reuse the masks prepared for the specialized models to evaluate the visual reconstruction metrics of our model in unedited regions. Based on the mask information, all regions not covered by the mask are defined as unedited areas. We measure PSNR, SSIM, MS-SSIM, and LPIPS in these regions as indicators of original video preservation. Figure 15 illustrates the evolution of these preservation metrics across different training stages.

From Figures 15 and 17, we observe that during the pretraining stage, where the pretraining data consists of video clips sampled from different temporal intervals, the model learns only basic editing concepts and the preservation of the visual appearance of various elements in the original video. However, due to the nature of the pretraining clip data, the model fails to effectively learn to preserve temporally consistent content in the unedited regions.

As illustrated in Figure 15, once temporally aligned editing pair data is incorporated for SFT, the performance of original video preservation improves rapidly. The model acquires strong temporal consistency in unedited regions after only a small number of training steps. Two factors contribute to this rapid acquisition. First, when the original video is encoded into latents via the VAE and fed to the model, these latents represent relatively shallow features that are easily learned. Second, the token sequence concatenation mechanism ensures that the noisy tokens are repeatedly exposed to the original video tokens, allowing them to fully extract information from the original video. Consequently, the ability to preserve original video content is learned very quickly.

**Editing Instruction Prompt Analysis.** We perform semantic clustering to analyze the editing instructions from the pretraining clip data, the SFT editing data, and the test data. Specifically, all instructions are first encoded into normalized semantic vectors using E5-large-v2 (Wang et al., 2022). The resulting embeddings are then clustered using K-Means (Lloyd, 1982) to examine the semantic distribution across different data sources. Finally, UMAP (McInnes et al., 2018) is applied to reduce the high-dimensional vectors to two dimensions, which enables the visualization of the overall distribution of editing instructions in the semantic space.

In Figure 16, the red points represent the editing instructions from the pretraining clip data. We observe that these instructions already cover the semantic regions corresponding to addition (blue), removal (green), and replacement (purple) data. The stylization instructions (orange) appear distinctively in the lower-right region and are not covered by the red points. This occurs because stylization instructions cannot be simulated using pretraining clip data. As discussed in the ablation study, although stylization is not modeled during the pretraining stage, the model rapidly acquires stylization

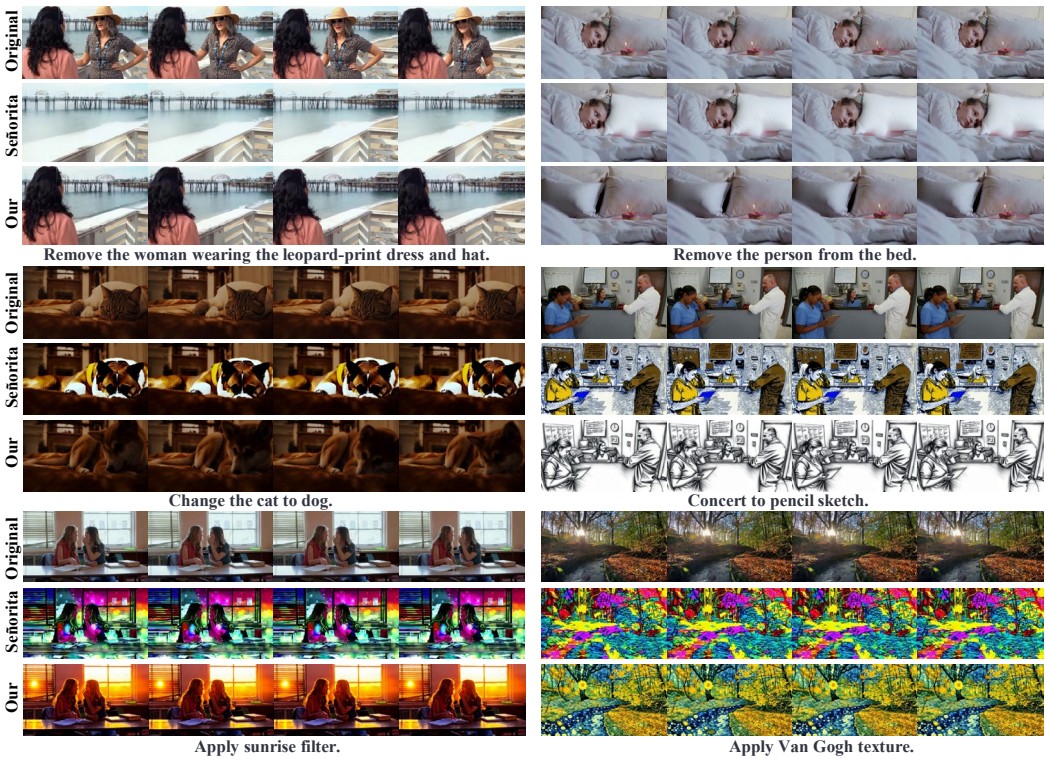

Figure 13: Ablation study on training data. The model trained on Señorita-2M (Zi et al., 2025b) exhibits limited editing capabilities, whereas the model trained on our curated data produces more natural and high-quality editing results.

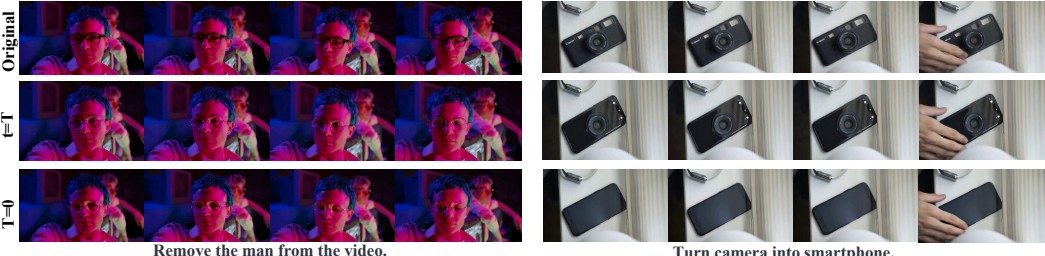

Figure 14: Ablation on timestep settings. When the timesteps for both the original video and the noisy video are set to the same value ($t = T$), we observe poorer preservation of details in the original video and the occurrence of copy-paste artifacts.

capabilities once SFT data is introduced. The cyan points represent the instructions from our test data, which are distributed uniformly across the semantic space, ensuring that the model's editing capability is evaluated comprehensively.

## F    MORE PRETRAINING RESULTS

Figure 17 presents several examples from the model pretrained solely on video clip data without any editing pairs. The results show that pretraining on clip data enables the model to acquire basic concepts of video editing. However, in the absence of explicit editing data, the model still produces noticeable errors, and the generated outputs do not strictly adhere to the given instructions. In our approach, pretraining is terminated at 40K steps, which provides a balance between learning basic editing concepts from clip data and avoiding model collapse.

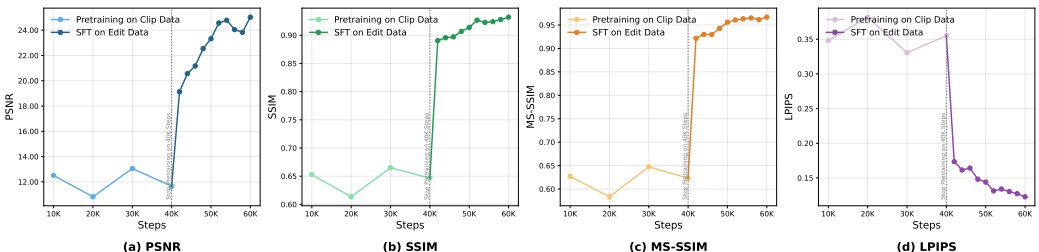

Figure 15: Model performance on the Original Video Preservation score across varying training steps. Once temporally aligned editing data is introduced for SFT (after 40K steps), the ability to preserve the original video is learned rapidly within a small number of additional training steps.

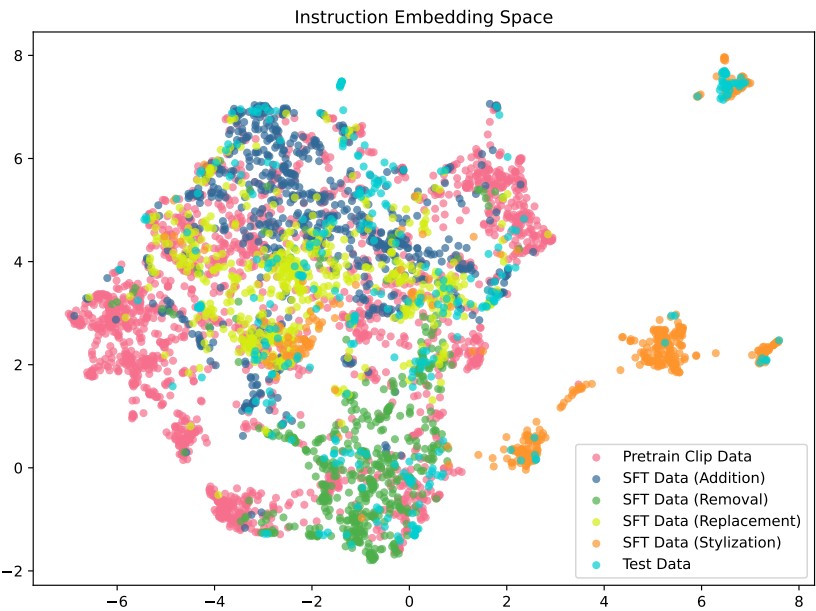

Figure 16: Semantic clustering of editing instructions. The instructions from the pretraining clip data already cover most types of editing operations, enabling the model to generalize addition, removal, and replacement behaviors during the pretraining stage.

## G MORE COMPARISON RESULTS

In Figure 18, 19, more qualitative comparisons are shown. Our proposed video editing model achieves more accurate editing instruction following and generates high-quality video results.

## H USER STUDY INTERFACE

Figure 20 presents the user study interface, where participants are required to rate the anonymized model outputs according to three evaluation criteria.

## I EDITING INSTRUCTION PROMPT

During the pretraining stage, Step3 (Wang et al., 2025a) is employed as the captioning model. The Table 6 shows the prompt used for generating editing instructions from video clip data.

## J    AUTOMATIC EVALUATION PROMPTS

In the experiments, GPT-5 (OpenAI, 2025) is used for the automated evaluation of editing metrics. Several representative prompts for guiding the large language model in performing automatic evaluation are presented in Tables 7, 8, 9, 10.

## K    LIMITATIONS AND FUTURE WORK

The sequence concatenation manner of the original and noised video tokens nearly doubles the sequence length, leading to quadratic computational costs in the full attention operation. However, much of the visual content within the original video tokens is redundant. In future work, we plan to explore effective approaches for compressing original video tokens and reducing the sequence length, thereby enabling more efficient instruction-based video editing.

## L    THE USE OF LARGE LANGUAGE MODELS

In this paper, large language models are primarily employed to correct grammatical errors and refine sentence structures, thereby improving the linguistic accuracy and clarity of expression, and helping the paper better conform to academic writing standards and enhance its overall readability.

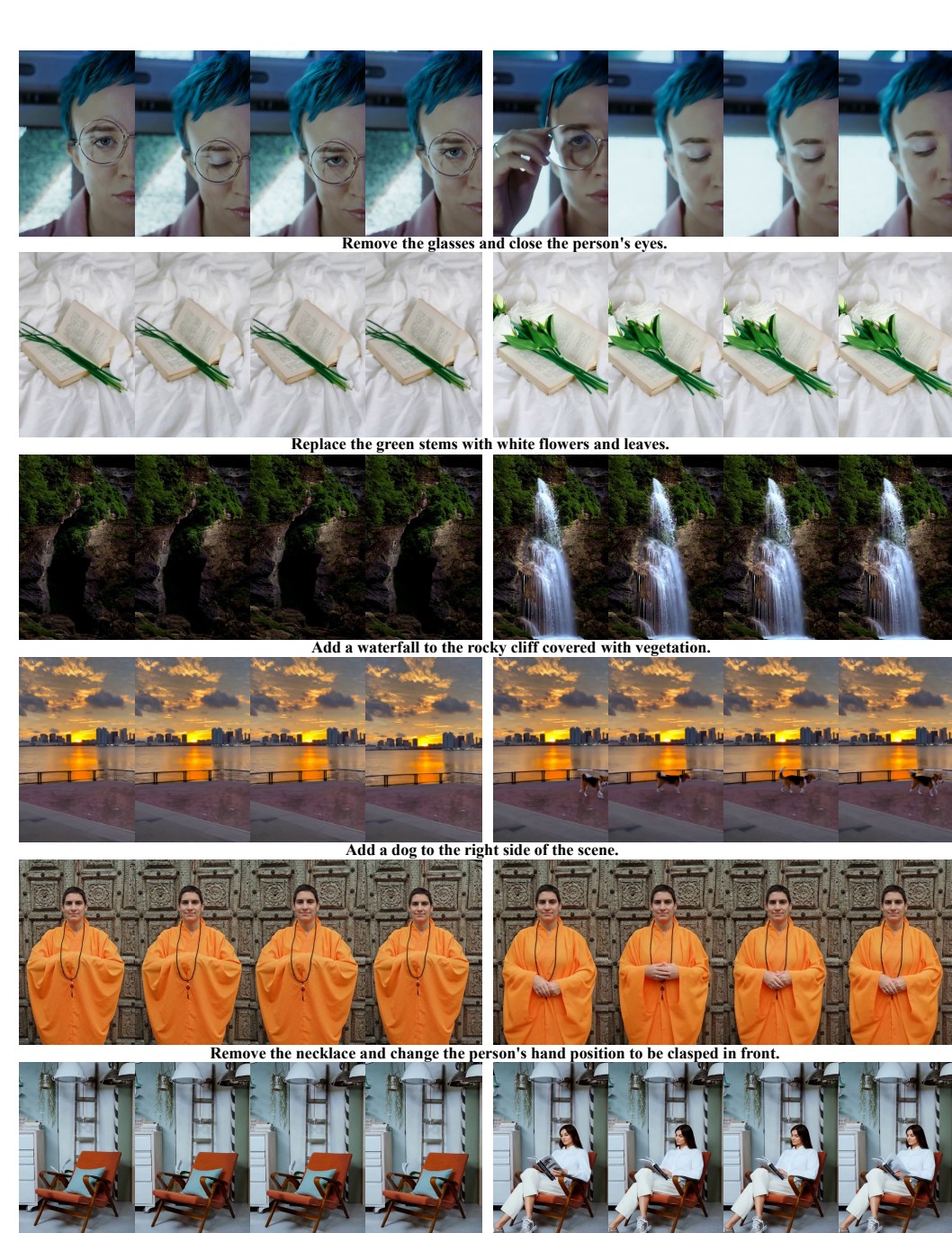

Remove the glasses and close the person's eyes.

Replace the green stems with white flowers and leaves.

Add a waterfall to the rocky cliff covered with vegetation.

Add a dog to the right side of the scene.

Remove the necklace and change the person's hand position to be clasped in front.

Replace the empty chair with a person sitting and reading a magazine.

Figure 17: Visualization results of the pretrained model. The results show that the model acquires basic editing operations using only pretraining clip data. The left figures are the original video, and the right figures are the edited results by the pretrained model.

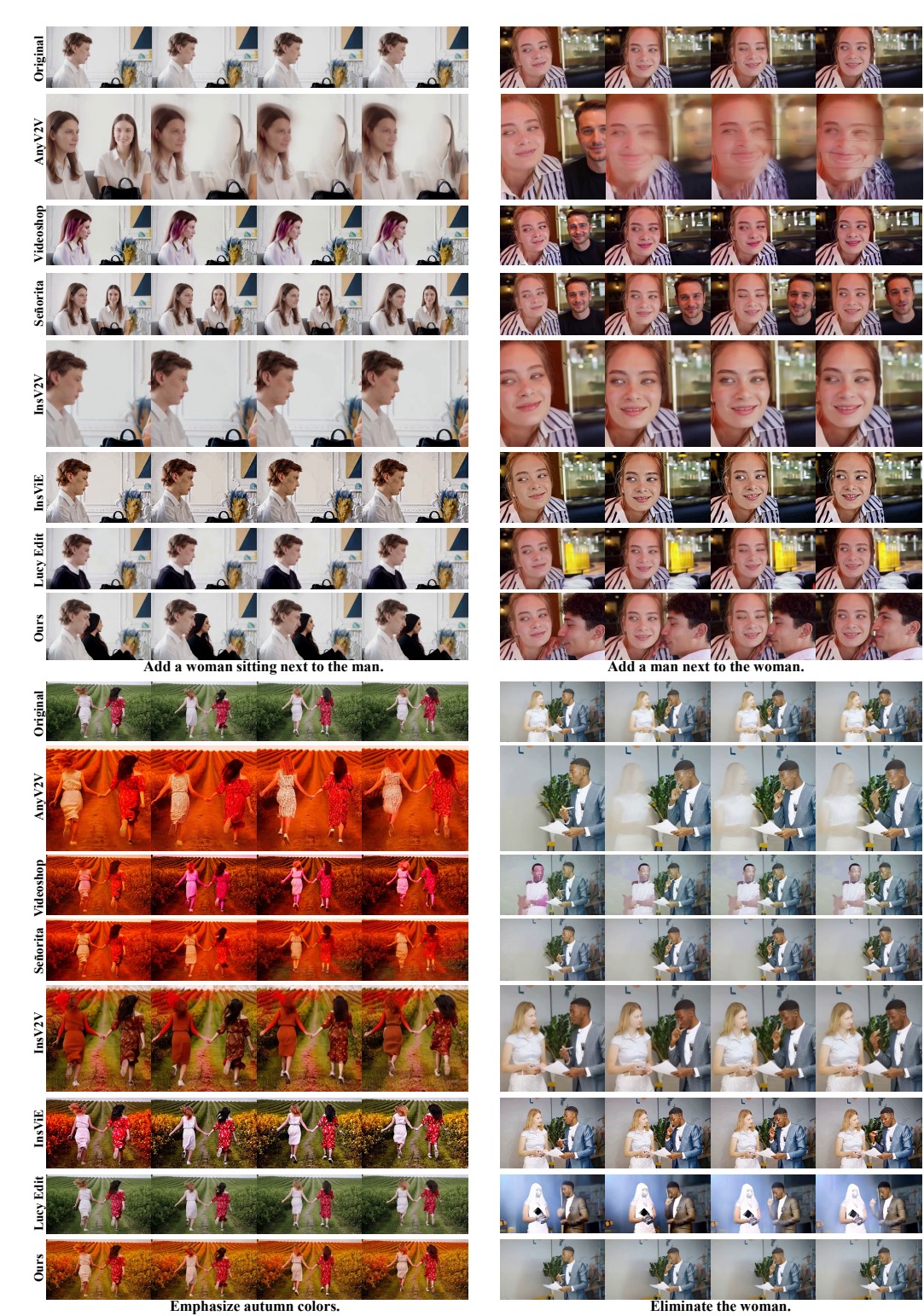

Figure 18: More comparison results with instruction-based methods. For methods requiring the first edited frame, the frame is generated using Qwen-Image-Edit (Wu et al., 2025a).

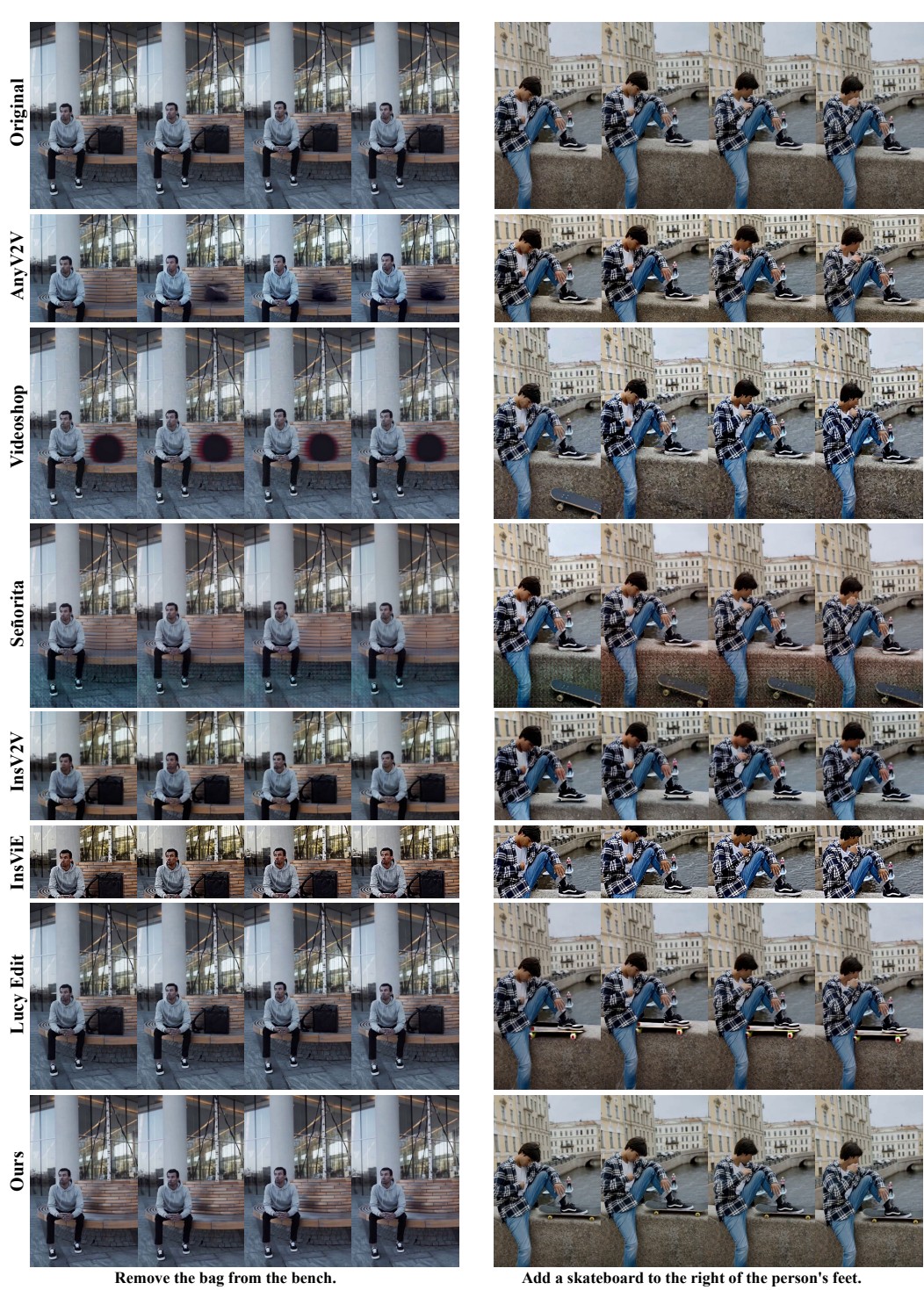

Figure 19: More comparison results with instruction-based methods. For methods requiring the first edited frame, the frame is generated using Qwen-Image-Edit (Wu et al., 2025a).

Figure 20: The interface of the user study.

Table 6: Prompt template for video editing instruction generation.

| **Prompt Template: Video Editing Instruction Generation** |
| --- |
| You are an advanced vision–language model tasked with generating precise video-editing instructions based on the original video and the edited video. |
| The following images are frames from the original video and the edited video: the left image shows the original frame, and the right image shows the edited frame. |
| You should generate a video-editing instruction describing the differences between the original and the edited version. |
| You must reply with only **one instruction representing the most significant editing operation, with no additional analysis text**. |
| The instruction should be concise and specific, focusing on the primary change made in the edited version. |
| Do not use words such as "frame", "image" or "video" in your response. |

Table 7: Prompt template for instruction following scoring.

| **Prompt Template: Instruction Following Scoring** |
|---|
| You are an advanced vision–language model tasked with grading instruction adherence for a video edit. I will provide several paired frames from an original video (left) and its edited version (right), separated by a blank divider. |
| The editing instruction is: `<instruct_prompt>` |
| Judge how well the edited video **follows the instruction only, without introducing unrelated edits**. Score on a 1–5 scale and output only one digit: |
| 5    perfectly follows; all required changes are present; no unrelated changes |
| 4    mostly follows; minor missing details or minor unrelated changes |
| 3    partially follows; noticeable missing parts or some unrelated changes |
| 2    poorly follows; major missing parts or many unrelated changes |
| 1    does not follow; requested edits absent or mostly wrong |
| Output only one digit: 1, 2, 3, 4, or 5. No words, no punctuation, no explanation. |

Table 8: Prompt template for original video preservation scoring.

| **Prompt Template: Original Video Preservation Scoring** |
|---|
| You are an advanced vision–language model acting as a video edit preservation rater. I will provide several sampled frame pairs from one original video (left) and its edited version (right), separated by a blank divider. |
| The editing instruction for this case is: `<instruct_prompt>` |
| Your task: |
|     1. From the instruction, infer which elements should be changed (the requested edits). |
|     2. For all other elements that are not requested to change (backgrounds, identities, layout, colors, etc.), **judge how well they are preserved in the edited video compared with the original**. |
| Rate only the preservation of the unedited parts on a 1–5 scale and output only one digit: |
| 5    unedited parts are preserved very well with minimal unintended changes |
| 4    mostly preserved; minor unintended changes |
| 3    partly preserved; noticeable unintended changes |
| 2    poorly preserved; major unintended changes |
| 1    not preserved; extensive unintended changes |
| Output only one digit: 1, 2, 3, 4, or 5. No words, no punctuation, no explanation. |

Table 9: Prompt template for editing quality scoring.

| **Prompt Template: Editing Quality Scoring** |
| --- |
| You are an advanced vision–language model acting as a video quality rater. I will provide several sampled frames from a single edited video. 

 Assess the overall **visual quality** considering the following aspects: 
 • sharpness and detail preservation 
 • noise, compression artifacts, or blocking 
 • color banding and gradient smoothness 
 • flicker or temporal stability (as inferred from frames) 
 • exposure, contrast, and color accuracy 
 • deformations or obvious editing artifacts 


 Rate the overall visual quality on a 1–5 scale and output only one digit: 

 5    excellent, very clear, minimal artifacts 
 4    good, only minor artifacts 
 3    fair, noticeable artifacts but acceptable 
 2    poor, strong artifacts, blur, or flicker 
 1    very poor, severe degradation 
 Output only one digit: 1, 2, 3, 4, or 5. No words, no punctuation, no explanation. |

Table 10: Prompt template for success ratio.

| **Prompt Template: Success Ratio** |
| --- |
| You are an advanced vision–language model tasked with verifying video edits. 

 The following images are sampled frames from the original and edited versions: the left sub-image shows the original, and the right sub-image shows the edited result. The two sub-images are separated by a blank divider. 

 The video-editing prompt for this case is: `<instruct_prompt>` 

 Your task is to: 

 1. understand the content of the original video from its frames, 
 2. examine the edited video frames and determine whether the changes match the editing prompt, 
 3. ensure that no additional edits are introduced—only the modifications required by the prompt are allowed. 


 You must reply with only a single lowercase word: `yes` or `no`. No explanations, punctuation, or extra text. |

