# OpenReview forum: "In-Context Learning with Unpaired Clips for Instruction-based Video Editing"
_ICLR.cc/2026/Conference — Submitted to ICLR 2026_

### Official Review · Reviewer_aSfy · 2025-10-17

**Soundness:** 3
**Presentation:** 3
**Contribution:** 3
**Rating:** 6
**Confidence:** 4

**Summary:**

This paper introduces a data-efficient training strategy for instruction-based video editing that reduces reliance on large paired datasets. The method first pretrains on unpaired video clips to learn general editing concepts, then fine-tunes on a small set of high-quality editing pairs. Built upon HunyuanVideoT2V, the approach achieves significant gains in instruction alignment and visual quality over existing methods (e.g., Senorita-2M, InsViE-1M), with 12% improvement in instruction following and 15% in editing quality. The paper includes detailed ablations demonstrating that pretraining on clip data provides strong editing priors and enables effective fine-tuning with limited paired data.

**Strengths:**

- The paper addresses an important and timely problem in instruction-based video editing, where collecting large-scale paired datasets is prohibitively expensive.
- The proposed idea of using unpaired video clips for pretraining is novel, practical, and conceptually simple, yet it leads to strong empirical improvements.
- The experiments are comprehensive and include comparisons with several baselines, detailed ablations, and qualitative results that convincingly demonstrate the method’s effectiveness.
- The paper is clearly written and well-structured, with strong visuals that help the reader understand the data curation pipeline and model design.

**Weaknesses:**

- The main innovation lies in the data strategy rather than the model architecture, which is only moderately modified from HunyuanVideoT2V.
- The evaluation relies heavily on automated metrics (such as CLIP similarity and GPT-5-based scoring), without human studies to verify perceptual quality or instruction alignment. Pairwise comparison with ELO rating would be helpful.
- The paper does not thoroughly analyze the computational cost of pretraining on one million clips, which could still be resource-intensive in practice.
- The generalization ability of the approach to other domains, such as stylized or synthetic videos, remains unclear and could have been explored further.

**Questions:**

1. How does the performance change when the model is pretrained on smaller subsets of clip data (for example, 100k or 200k clips)?
2. Could the same pretraining and fine-tuning strategy be applied to other backbones beyond HunyuanVideoT2V?
3. Have you evaluated how well the model performs on non-natural video domains such as animation, cartoons, or synthetic datasets?
4. Could the authors provide an estimate of the compute or training time required for the 1M-clip pretraining stage?

---

> ### Author Response · Authors · 2025-11-27
>
> Thank you for your efforts to help improve the quality of our manuscript. We have carefully revised the paper according to your suggestions (newly added content is marked in blue). Regarding the weaknesses and questions you raised, we summarize them into the following seven concerns and address them below.
>
> ---
>
> **Concern 1: Moderate model modification.**
>
> The token sequence concatenation mechanism enables the model to effectively learn information from the original video, while our modification of the timestep $t$ is introduced to maintain generation quality. These lightweight architectural changes preserve the generative capability of the underlying T2V foundation model.
>
> ---
>
> **Concern 2: Lack of user study.**
>
> We conduct a user study to evaluate the editing quality of our method; please refer to Section 4.3 of the main body. Our method consistently outperforms other instruction-based approaches across all evaluation dimensions.
>
> ---
>
> **Concern 3: Analysis of computational resources.**
>
> As detailed in Section C of the supplementary material, we provide comprehensive details regarding the computational resources required for both data preparation and model training.
>
> ---
>
> **Concern 4: Out-of-Distribution test data.**
>
> As presented in Section D of the supplementary material (Results on Out-of-Distribution Data), we construct an out-of-distribution evaluation set and report the corresponding results. Our method demonstrates the ability to perform editing on these out-of-distribution videos.
>
> ---
>
> **Concern 5: Necessity of sufficient pretraining data.**
>
> As shown in Section 4.4 of the main body, we conduct an ablation study using a reduced amount of clip data for pretraining. This setting results in noticeable artifacts and unnatural generation outputs, demonstrating that insufficient pretraining data degrades the model's generative quality. Combined with the findings from the ablation on training strategies, these results indicate that a large quantity of pretraining data is essential, regardless of whether the model is trained using our proposed strategy or via a paradigm that relies entirely on editing data. Our method is specifically designed to alleviate the need for large-scale pretraining editing datasets by using clip pairs, which provide pretraining data at low cost.
>
> ---
>
> **Concern 6: Other backbone.**
>
> As shown in Section E of the supplementary material (Ablation on Foundation Models), we also replicate our editing model on another video generation backbone, Wan 2.1 14B [1], and report consistent results.
>
> ---
>
> **Concern 7: Pretraining time and computational cost.**
>
> As detailed in Section C of the supplementary material, we provide the required computational resources and the total training time for the pretraining stage.
>
> ---
>
> **References**
>
> [1] Wan T, Wang A, Ai B, et al. Wan: Open and advanced large-scale video generative models. arXiv preprint arXiv:2503.20314, 2025.

---

### Official Review · Reviewer_86hF · 2025-10-25

**Soundness:** 2
**Presentation:** 3
**Contribution:** 2
**Rating:** 4
**Confidence:** 3

**Summary:**

This paper addresses the challenge of data scarcity for instruction-based video editing by proposing a novel two-stage training strategy. The approach first pretrains a foundation video generation model (based on Hunyuan VideoT2V) using In-Context Learning (ICL) with unpaired video clips. The pretraining stage leverages approximately 1 million real video clips, treating clips sampled from the same scene segment but different temporal intervals as pseudo-original and pseudo-edited pairs. An instruction is automatically generated to describe the difference between the two clips. This stage teaches the model basic editing concepts (e.g., addition, replacement, deletion) and strengthens its ability to preserve original video content. The second stage involves Supervised Fine-Tuning (SFT) on a small, high-quality synthetic dataset of fewer than 150k curated editing pairs to extend editing tasks and improve quality. The model architecture uses an in-context approach where the original video tokens (with timestep $t=0$) are concatenated with the noised video tokens. The proposed method achieves superior performance against existing instruction-based video editing models, with reported improvements of 12% in instruction following and 15% in editing quality.

**Strengths:**

1. The combination of large-scale, low-cost pretraining on real-world clips (learning basic concepts and preservation) followed by targeted SFT on a small, high-quality synthetic set (learning complex edits) is highly effective and data-efficient.
2. The method achieves state-of-the-art results, showing significant gains (12% and 15%) in instruction following and editing quality compared to existing methods.
3.  The model achieves superior results using only $\sim$1M video clips and $<150k$ paired editing samples, a fraction of the data required by comparable SOTA models.

**Weaknesses:**

1. The data curation process heavily relies on powerful external models (Step3 for instruction generation/filtering, GroundedSAM2 for masking, VACE for inpainting, Qwen2.5-VL for filtering). This dependency raises questions about the generalizability of the pipeline if these auxiliary models change or are unavailable.
2. While the ablation study is mentioned, more granular detail on the performance difference between: a) No pretraining + SFT, b) Pretraining only, and c) Full two-stage training is crucial to quantify the specific gain from the ICL pretraining stage. (The current abstract only mentions the final performance ).
3. Built upon Hunyuan VideoT2V, the model size and hardware requirements are implicitly high. A brief discussion on the computational resources needed for training (not just data generation) compared to other SOTA models would provide a more complete picture of the "low-cost" claim.

**Questions:**

1. Can the authors characterize the instructions generated from the unpaired clips (e.g., using a distribution plot or semantic clustering) and compare their nature (e.g., motion, camera work, lighting changes) to the instructions in the SFT and testing datasets? This would help justify the claim that these "basic editing concepts" are effectively transferable.
2. Given the heavy reliance on an in-context token concatenation approach, how is temporal consistency explicitly addressed beyond the base DiT architecture? Did the authors find that the separation of original and noised tokens (via $t=0$ and $t=T$) impacts the temporal self-attention in the DiT blocks, and if so, how was this mitigated?
3. The SFT uses "fewer than 150k" samples for only "one epoch". This is an extremely low training budget. Please confirm if the entire Hunyuan VideoT2V backbone is updated during SFT, or if only a subset of parameters (e.g., attention layers or a LoRA adapter) is fine-tuned. This detail is crucial for assessing the true efficiency.

---

> ### Author Response · Authors · 2025-11-27
>
> Thank you for your efforts to help improve the quality of our manuscript. We have carefully revised the paper according to your suggestions (newly added content is marked in blue). Regarding the weaknesses and questions you raised, we summarize them into the following five concerns and address them below.
>
> ---
>
> **Concern 1: Generalization of the data curation pipeline.**
>
> Our data curation pipeline is designed to be modular and easily generalizable. The components used in the current implementation can be replaced by alternative models that serve similar functions. For example, Step3 and Qwen2.5-VL can be substituted with closed-source models such as ChatGPT-5 or Gemini-3, or with other open-source VLMs to further improve annotation quality. Likewise, the inpainting module can be replaced with other video inpainting approaches, such as Minimax-Remover [1] or VideoPainter [2]. The detection and video segmentation models used in the pipeline can also be swapped with equivalent methods. This flexibility demonstrates that our pipeline is not tied to a specific set of models and can be readily adapted to different tool choices and resource availability.
>
> ---
>
> **Concern 2: More details on the different training settings.**
>
> As discussed in Section 4.4 of the main body, we provide a detailed analysis of how the instruction following score evolves throughout the training process. The clip data pretraining stage equips the model with basic editing capabilities and ensures high-quality video generation. This strong initialization enables the model to achieve superior final performance after only a small amount of SFT with high-quality editing data. In contrast, the other two training settings either lack sufficient pretraining or rely solely on synthetic editing pairs, resulting in lower overall performance.
>
> ---
>
> **Concern 3: Discussion about computational resources.**
>
> As detailed in Section C of the supplementary material, we provide a comprehensive description of the computational resources used during both the data preparation and the model training process.
>
> ---
>
> **Concern 4: Instruction prompt analysis.**
>
> As presented in Section E of the supplementary material (Editing Instruction Prompt Analysis), we conduct a semantic clustering analysis of the instruction prompts across different data sources. Based on the resulting clusters, we analyze why our method effectively transfers editing capabilities from clip data pretraining to editing data during SFT. The clustering results indicate that the semantic distribution of the clip data aligns well with most of the editing categories, thereby enabling effective capability transfer in the SFT stage.
>
> ---
>
> **Concern 5: Temporal consistency.**
>
> In our design, the original video and the nosied video share the same aligned positional encoding. This allows the model to leverage the spatiotemporal understanding already learned by the underlying T2V foundation model, enabling it to recognize the spatiotemporal correspondence between the original and noisy video tokens. Setting t = 0 for the original video does not affect temporal consistency. Instead, this modification preserves the foundational assumption of the T2V model that t = 0 corresponds to a noise-free input.
>
> ---
>
> **Concern 6: Training setting.**
>
> As described in Section A of the supplementary material, our model is fully fine-tuned during both the pretraining and SFT stages.
>
> ---
>
> **References**
>
> [1] Zi B, Peng W, Qi X, et al. MiniMax-Remover: Taming Bad Noise Helps Video Object Removal. arXiv preprint arXiv:2505.24873, 2025.
>
> [2] Bian Y, Zhang Z, Ju X, et al. Videopainter: Any-length video inpainting and editing with plug-and-play context control. Proceedings of the Special Interest Group on Computer Graphics and Interactive Techniques Conference Conference Papers. 2025: 1-12.

---

### Official Review · Reviewer_T1N6 · 2025-10-28

**Soundness:** 2
**Presentation:** 3
**Contribution:** 2
**Rating:** 4
**Confidence:** 5

**Summary:**

This paper presents an "in-context learning" approach for training an instructional video editing model through a two-stage paradigm: unpaired pre-training followed by paired supervised fine-tuning. Compared to existing methods, the proposed approach achieves reasonable performance improvements. While I acknowledge the systematic effort invested in large-scale data processing and model training, the work currently offers limited novel insights or technical contributions beyond the engineering effort. I consider this a borderline paper at this stage and look forward to the authors' response.

**Strengths:**

1. The proposed pre-training stage on unpaired clip data mitigates the data scarcity issue inherent in instructional video editing, where obtaining strictly paired data is challenging. This approach enables the model to establish fundamental instructional editing capabilities even without strict pairing, which is a practical and valuable contribution.

2. Through large-scale pre-training followed by supervised fine-tuning (SFT), the proposed model demonstrates consistent and reasonable performance improvements on instructional video editing tasks compared to existing baselines.

**Weaknesses:**

1. While I acknowledge that pre-training on large-scale unpaired data helps the model shift from understanding descriptive text to instructional text, which benefits instructional comprehension, I'm concerned about its impact on video preservation. Since unpaired data typically contains videos with significant differences where very few elements are strictly preserved, this pre-training stage may harm the model's ability to preserve the original video content—an important aspect of video editing. I would suggest including more automatic metrics in the ablation study to evaluate video preservation under different training paradigms, rather than relying solely on the O_P score from GPT-5, which would strengthen the analysis.

2. I'm not convinced that the term "in-context learning" is appropriate here, given that the original video is essentially used as a condition through sequence concatenation. Could the authors elaborate on why this framing is justified?

3. I'm curious about the design choice of setting t=0 for the original video tokens during training. Intuitively, both t=0 and t=T seem viable. I understand this design might help the model distinguish the original video from noisy frames, but do the authors have experimental results comparing these choices? How much does this specific design contribute to performance, and what's the deeper reasoning behind it?

4. What is the SFT dataset used in this paper? I couldn't find detailed information about it in the manuscript.

**Questions:**

See wekanesses.

---

> ### Author Response · Authors · 2025-11-27
>
> Thank you for your efforts to help improve the quality of our manuscript. We have carefully revised the paper according to your suggestions (newly added content is marked in blue). Regarding the weaknesses and questions you raised, we summarize them into the following four concerns and address them below.
>
> ---
>
> **Concern 1: Automatic evaluation for video preservation.**
>
> As detailed in Section E of the supplementary material (Performance of Original Video Preservation), we employ standard image reconstruction metrics, including PSNR, SSIM, and LPIPS, to automatically evaluate video preservation quality. Our ablation analysis shows that the model rapidly acquires a strong video preservation capability once SFT data is introduced. The degradation of video preservation introduced during the pretraining stage is effectively corrected after only a small number of SFT steps.
>
> ---
>
> **Concern 2: Definition of in-context learning.**
>
> In our paper, "in-context" refers to the relationship between the two video clips used in pretraining, where the two clips share similar scenes and contain largely the same visual elements. "Learning" refers to capturing this in-context relationship during pretraining to endow the model with basic editing capabilities. The term "in-context learning" describes the pretraining stage in which the model learns basic editing concepts, such as changing some elements while preserving others according to the editing instruction, from paired video clips.
>
> There is also a definition regarding the "in-context condition", which describes the mechanism of condition injection, such as the original video, by concatenating condition tokens with noised video tokens along the token sequence, following the definition of Scalable Diffusion Models with Transformers [1]. To avoid misunderstanding, we have revised the terminology used in describing the model architecture and replaced it with the more precise expression "token sequence concatenation".
>
> ---
>
> **Concern 3: Setting t = 0 for the original video.**
>
> This design choice serves two purposes. First, the original video is provided to the model without any added noise, where t = 0 corresponds to a noise-free input in the setting of the original T2V foundation model. This modification helps maintain the generative capability of the original T2V backbone and ensures high-quality video synthesis. Second, setting t = 0 allows the model to clearly distinguish between the original video and the noisy video. In our architecture, both the original and the nosied videos pass through the same normalization module. Assigning t = 0 for the original video ensures that the normalized distributions of the two types of tokens are different, enabling the model to identify the original video tokens from the noised video.
>
> We present the ablation study results in Section E of the supplementary material (Ablation on Timestep for Original Video). When the timesteps for both the original video and the noised video are set to the same value (t=T), we observe poorer preservation of details in the original video and the occurrence of copy-paste artifacts.
>
> ---
>
> **Concern 4: Curation of the SFT data.**
>
> As described in Section 3.1 of the main paper, the SFT data is curated from two sources. A portion of the data is generated using our data pipeline with videos sourced from the Pexels website, while the remaining portion is obtained by filtering existing publicly available datasets.
>
> ---
>
> **References**
>
> [1] Peebles W, Xie S. Scalable diffusion models with transformers. Proceedings of the IEEE/CVF international conference on computer vision. 2023: 4195-4205.

---

### Official Review · Reviewer_WZm9 · 2025-11-01

**Soundness:** 2
**Presentation:** 3
**Contribution:** 2
**Rating:** 6
**Confidence:** 3

**Summary:**

In this paper, it is demonstrated that pretraining a foundation model on unpaired video clips, followed by fine-tuning on supervised fine-tuning (SFT) data, can significantly improve the model’s performance. Using HunyuanVideoT2V as the base framework, the model is first pretrained on approximately one million real video clips to acquire general video editing capabilities, and then fine-tuned on fewer than 150K carefully curated editing pairs to further enhance performance and broaden editing abilities. The experimental results show strong performance and provide comprehensive comparisons with previous approaches.

**Strengths:**

1. The paper effectively leverages unpaired video–text clips to train a model that can be easily fine-tuned into an instruction-based framework, demonstrating flexibility and scalability.

2. The editing results are visually impressive and show clear improvements in realism and semantic consistency.

3. Compared with existing video editing approaches, the proposed model achieves competitive or superior performance across several qualitative and quantitative evaluations, highlighting its potential practical value.

**Weaknesses:**

1. Some fine-grained details are lost after editing. For example, in Figure 6, the hair in the first row becomes noticeably blurred, indicating a limitation in preserving texture details.

2. The model occasionally fails to fully follow the given instructions. In Figure 6, fifth row, the necklace remains visible despite the instruction to remove it, suggesting incomplete semantic alignment.

3. The model has not been evaluated on alternative architectures such as WAN2.1 1.3B, so the generalization capability of the proposed method across different backbones remains unclear.

4. The paper does not include comparisons with specialized models designed for specific editing purposes, such as Style Master or Minimax-Remover, which could provide a more comprehensive evaluation.

5. The ablation study is insufficient. It would be valuable to clarify whether, under identical training steps, a model based on Senorita would yield similar editing performance, helping to verify the true contribution of the proposed method.

**Questions:**

Please see the Weaknesses.

---

> ### Author Response · Authors · 2025-11-27
>
> We thank the reviewers for their constructive feedback, which helps improve the quality of our manuscript. We have carefully revised the paper according to your suggestions (newly added content is marked in blue). We summarize the weaknesses and questions raised into the following four concerns and address them below.
>
> ---
>
> **Concern 1: Degradation of editing performance after clip data pretraining.**
>
> Since the model is pretrained on video clip pairs extracted from different temporal intervals without access to temporally aligned editing data, the pretrained model occasionally generates a continuation of the video content rather than an edited result. As illustrated by the example in the first row, the generated results tend to continue the temporal dynamics and camera motion of the previous clip. This bias leads to potential issues such as unfocused shots and noticeably blurred hair, as seen in the first row of Figure 16. In the fifth example of Figure 16, the pretrained model fails to execute the editing instruction perfectly. This occurs primarily because the pseudo editing pairs used during pretraining rely solely on VLM-generated labels without filtering, which may contain inaccuracies. These labeling inaccuracies cause the model to fail to follow editing instructions in certain cases.
>
> We fully acknowledge that the editing outputs after clip data pretraining alone are not yet ideal. This is why the SFT stage with high-quality editing data is required. As demonstrated in our ablation study, these issues are effectively corrected after only a small number of SFT steps. The purpose of pretraining is not to achieve perfect editing, but rather to enable the model to: (i) learn to preserve the appearance and visual characteristics of objects in the original video; (ii) maintain high-quality generative results; and (iii) acquire basic editing capabilities. The SFT stage subsequently refines and aligns this capability for accurate instruction following.
>
> ---
>
> **Concern 2: Evaluation on an alternative foundation model.**
>
> As detailed in Section E of the supplementary material (Ablation on Foundation Models), we reproduce our video editing model on Wan 2.1 14B [1] using the same training strategy, data pipeline, and architectural modifications. The reproduced model demonstrates consistent behaviors and confirms that our method generalizes well across different video generation backbones.
>
> ---
>
> **Concern 3: Comparison with specific editing models.**
>
> As discussed in Section D of the supplementary material (Compared with Specialized Models), we compare our method with several specialized editing models, including VACE [2], Minimax-Remover [3], and Ditto [4]. Since StyleMaster [5] only releases its T2V weights and code and does not provide a video editing model, we select Ditto as the alternative method for stylization editing. Our method surpasses the specialized models in several task-specific editing scenarios. The instruction-based paradigm effectively avoids the strong dependence on accurate masks required by mask-based methods.
>
> ---
>
> **Concern 4: Ablation on training data.**
>
> As shown in Section E of the supplementary material (Ablation on Curated Data), we provide results obtained by training with the Señorita-2M [6] dataset under the same training configuration. The model trained with our data produces more natural and higher-quality video editing results, which further validates the advantages of our curated data pipeline and training strategy.
>
> ---
>
> **References**
>
> [1] Wan T, Wang A, Ai B, et al. Wan: Open and advanced large-scale video generative models. arXiv preprint arXiv:2503.20314, 2025.
>
> [2] Jiang Z, Han Z, Mao C, et al. Vace: All-in-one video creation and editing. arXiv preprint arXiv:2503.07598, 2025.
>
> [3] Zi B, Peng W, Qi X, et al. MiniMax-Remover: Taming Bad Noise Helps Video Object Removal. arXiv preprint arXiv:2505.24873, 2025.
>
> [4] Bai Q, Wang Q, Ouyang H, et al. Scaling Instruction-Based Video Editing with a High-Quality Synthetic Dataset. arXiv preprint arXiv:2510.15742, 2025.
>
> [5] Ye Z, Huang H, Wang X, et al. Stylemaster: Stylize your video with artistic generation and translation. Proceedings of the Computer Vision and Pattern Recognition Conference. 2025: 2630-2640.
>
> [6] Zi B, Ruan P, Chen M, et al. Señorita-2M: A High-Quality Instruction-based Dataset for General Video Editing by Video Specialists. arXiv preprint arXiv:2502.06734, 2025.

---

### Author Response · Authors · 2025-12-01
**Summary for Area Chairs**

**Note to the Area Chairs**

We understand that this year presents unique challenges, and we deeply appreciate the Area Chairs taking on the unprecedented workload of managing the rebuttal and discussion phase under these circumstances. We thank you for your time and effort in evaluating our work.

---

**Highlights from Reviews**

Reviewers consistently recognize our method as a practical, effective, and data-efficient solution for instruction-based video editing. They highlight the **“novel two-stage training strategy”** (86hF), noting that pretraining on unpaired clips is **“practical and valuable”** (T1N6) as well as **“novel, practical, and conceptually simple”** (aSfy), while remaining **flexible and scalable** (WZm9). Multiple reviewers affirm our **state-of-the-art performance**, citing **“12% improvement in instruction following and 15% in editing quality”** (86hF, aSfy) and describing our results as **“visually impressive”** with stronger realism and semantic consistency (WZm9). They further praise the data efficiency of achieving such performance with only **1M clip data and <150K SFT pairs**, calling the approach **“highly effective and data-efficient”** (86hF) with **“strong empirical improvements”** supported by **comprehensive experiments** (aSfy).

---

**Addressing Questions and Concerns**

Across 4 reviews, reviewers raised 15 distinct questions regarding evaluation scope, algorithmic design choices, and computational efficiency. We resolved 6 of these points through detailed clarification in our rebuttals, addressed the remaining 9 concerns by conducting extensive new experiments and analyses, and incorporated 11 key additions into the manuscript.

---

**Key Manuscript Changes**

All changes are highlighted in blue in the updated PDF. Key manuscript additions:

* **User Study (Section 4.3):** To verify perceptual quality, we conducted a human evaluation with 50 participants rating 20 videos across three dimensions. Results confirm our method significantly outperforms baselines in instruction following and visual fidelity.

* **Pretraining Data Scaling & Training Strategy (Section 4.4):** We expanded the ablation study to analyze the impact of pretraining data scale (100k vs 1M clips) and training steps. This validates that sufficient pretraining on large-scale clip data is essential for maintaining generative quality and preventing artifacts. Additionally, we evaluated our approach against standard training paradigms, including pretraining on synthetic editing data and direct fine-tuning without pretraining. The results demonstrate that our two-stage strategy effectively prevents the model collapse and artifacts associated with training solely on synthetic editing pairs, yielding superior performance with higher data efficiency.

* **Computational Resources (Appendix C):** We added a detailed statement of the hardware usage and time costs for both the data curation pipeline and the two-stage model training.

* **Comparison with Specialized Models (Appendix D):** We included comparisons against task-specific models. Results demonstrate our unified model achieves superior performance across several editing tasks.

* **Out-of-Distribution Generalization (Appendix D):** We added qualitative results on anime and synthetic videos. This demonstrates the model's robustness and capability to generalize to out-of-distribution domains not seen during training.

* **Foundation Model Generalization (Appendix E):** We validated the generalization of our strategy by re-implementing the method on the Wan 2.1 14B backbone. The consistent improvements confirm that our approach generalizes well across different foundation video generation models.

* **Data Pipeline Effectiveness (Appendix E):** We added an ablation training solely on the Señorita-2M dataset. The superior performance of our model confirms the advantages of our curated pretraining and SFT data pipeline over existing large-scale synthetic datasets.

* **Video Preservation Analysis (Appendix E):** We introduced automated metrics evaluated on unedited regions. The analysis quantitatively proves that the model rapidly acquires strong content preservation capabilities after the introduction of SFT data.

* **Timestep Configuration Ablation (Appendix E):** We conducted experiments comparing distinct ($t=0$ vs $t=T$) and identical timesteps. Results verify that setting $t=0$ for original video tokens is critical for preserving fine details and preventing copy-paste artifacts.

* **Instruction Prompt Analysis (Appendix E):** We performed semantic clustering on instructions from different data sources. The visualization confirms that pretraining clip data effectively covers the semantic space of basic editing operations, facilitating transfer during SFT.

---

### Meta-Review · Area_Chair_pLk2 · 2025-12-31

**Summary:**

The paper proposes a low-cost pretraining strategy for instruction-based video editing, leveraging in-context learning from unpaired video clips combined with limited paired data.

Reviewers generally recognize the practicality and data efficiency of the proposed two-stage training strategy. However, several concerns remain unaddressed or insufficiently resolved. Firstly, although the authors mentioned that SFT corrects some issues, the problem of fine-grained detail loss (e.g., blurred hair) and occasional incomplete instruction following (e.g., failure to remove a necklace) is not fundamentally resolved, as these limitations persist in some editing results. Secondly, while the authors revised the terminology to avoid misunderstanding, the justification for framing the approach as "in-context learning" still lacks sufficient theoretical depth to fully convince the reviewer. Thirdly, Despite the authors' claim that the data curation pipeline is modular, the heavy reliance on specific external models still raises questions about its generalizability in scenarios where these auxiliary models are unavailable or modified. Additionally, while the authors provided some training process analysis, the granular comparison between different training settings (no pretraining + SFT, pretraining only, full two-stage training) to quantify the specific gain from ICL pretraining is still insufficient. Besides, although the authors added a user study and out-of-distribution generalization results, the main innovation remains focused on the data strategy rather than model architecture modifications, which is a limitation noted by the reviewer. Furthermore, an ethics issue was raised by the Area Chair regarding the copyright and privacy of the curated video data, which requires further review.

Based on the above considerations, I think this manuscript does not match the ICLR’s requirement and I do not recommend to accept this manuscript.

**Reviewer Concerns:**

Concerns from WZm9 about the problem of fine-grained detail loss (e.g., blurred hair) and occasional incomplete instruction following (e.g., failure to remove a necklace) is not fundamentally resolved, as these limitations persist in some editing results.Despite the authors' claim that the data curation pipeline is modular, the heavy reliance on specific external models still raises questions about its generalizability in scenarios where these auxiliary models are unavailable or modified. While the authors provided some training process analysis, the granular comparison between different training settings (no pretraining + SFT, pretraining only, full two-stage training) to quantify the specific gain from ICL pretraining is still insufficient. Besides, although the authors added a user study and out-of-distribution generalization results, the main innovation remains focused on the data strategy rather than model architecture modifications.

**Reviewer Scores:**

Reviewers would keep their scores.

---

### Decision · Program_Chairs · 2026-01-26

Reject